# Learning Where and When to Reason in Neuro-Symbolic Inference

**Cristina Cornelio[†], Jan Stuehmer[∗‡], Shell Xu Hu[†], Timothy Hospedales[†]**
[†]Samsung AI, Cambridge
{c.cornelio,shell.hu,t.hospedales}@samsung.com
[‡]Heidelberg Institute for Theoretical Studies and Karlsruhe Institute of Technology
{jan.stuehmer}@h-its.org.com

## Abstract

The integration of hard constraints on neural network outputs is a very desirable capability. This allows to instill trust in AI by guaranteeing the sanity of that neural network predictions with respect to domain knowledge. Recently, this topic has received a lot of attention. However, all the existing methods usually either impose the constraints in a "weak" form at training time, with no guarantees at inference, or fail to provide a general framework that supports different tasks and constraint types. We tackle this open problem from a neuro-symbolic perspective. Our pipeline enhances a conventional neural predictor with (1) a symbolic reasoning module capable of correcting structured prediction errors and (2) a neural attention module that learns to direct the reasoning effort to focus on potential prediction errors, while keeping other outputs unchanged. This framework provides an appealing trade-off between the efficiency of constraint-free neural inference and the prohibitive cost of exhaustive reasoning at inference time. We show that our method outperforms the state of the art on visual-Sudoku, and can also benefit visual scene graph prediction. Furthermore, it can improve the performance of existing neuro-symbolic systems that lack our explicit reasoning during inference.

## 1 Introduction

Despite the rapid advance of machine learning (ML), it is still difficult for deep learning architectures to solve a certain classes of problems, especially those that require non-trivial symbolic reasoning (e.g. automated theorem proving or scientific discovery). A very practical example of this limitation – even in applications that are typical deep learning territory such as image processing – is the difficulty of imposing hard symbolic constraints on model outputs. This is relevant when learning systems produce outputs for which domain knowledge constraints apply (e.g., Figure 2). The common situation today, that ML systems violate such constraints regularly, is both a missed opportunity to improve performance and more importantly a source of reduced public trust in AI.

This issue has motivated a growing body of work in neuro-symbolic methods that aim to exploit domain knowledge constraints and reasoning to improve performance. Most of these methods address neuro-symbolic *learning*, where constraints are applied in the loss function (e.g., Xu et al. (2018); Xie et al. (2019); Li et al. (2019); Wang & Pan (2020)) and predictions that violate those constraints are penalised. In this way, during learning, the model is "encouraged" to move close to a solution that satisfies the constraints/rules. However, high-capacity deep networks in any case usually fit their training sets, and thus violate no constraints on the output labels during supervised learning. The central issue of whether constraints are also met upon inference during deployment is unaddressed by these methods and is under-studied more generally Giunchiglia et al. (2022b); Dash et al. (2022); von Rueden et al. (2021). A minority of studies have worked towards exploiting constraints during inference. Since in general reasoning to guarantee that constraints are met is expensive, some methods try to apply soft relaxations (Daniele & Serafini, 2019; Li & Srikumar, 2019; Wang et al., 2019), which is unhelpful for trust and guarantees. The few methods that manage to impose exact constraints are either restricted to very simple or restrictive rules (Yu et al., 2017; Giunchiglia et al.,

---

[∗]Work done while at Samsung AI, Cambridge

2022b) that are not expressive enough, or involve invoking a full reasoning engine during inference Manhaeve et al. (2018); Yang et al. (2020), which is prohibitively costly in general.

In this work we explore a new neuro-symbolic integration approach to manage the trade-off between the cost, expressivity, and exactness of reasoning during inference. Our Neural Attention for Symbolic Reasoning (NASR) pipeline leverages the best of neural and symbolic worlds. Rather than performing inexact, inexpressive, or intractable reasoning, we first execute an efficient neural-solver to solve a task, and then delegate a symbolic solver to correct any mistakes of the neural solver. By reasoning over only a subset of neural fact predictions we maintain efficiency. So long as the facts selected have no false positives, we guarantee constraints are met during inference. Thus we enjoy most of the benefit of symbolic reasoning about the solution with a cost similar to neural inference. This dual approach is aligned with the 'two systems' perspective of Sloman (1996) and Kahneman (2011) that has recently been applied in AI (e.g. Booch et al. (2021) and LeCun (2022)). More specifically, our NASR framework is built upon three components: A neural network (Neuro-Solver) is trained to solve a given task directly; a symbolic engine that reasons over the output of the Neuro-Solver, and can revise its predictions in accordance with domain knowledge; and a hard attention neural network (Mask-Predictor), that decides which subset of Neuro-Solver predictions should be eligible for revision by the reasoning engine using domain knowledge. The Mask-Predictor essentially learns *when and where to reason* in order to effectively achieve high prediction accuracy and constraint satisfaction with low computation cost. Since the reasoning engine is not generally differentiable, we train our framework with reinforcement learning (RL).

The contributions of our work can be summarized as follows: (1) We provide a novel neuro-symbolic integration pipeline with a novel neural-attention module (Mask-Predictor) that works with any type of constraints/rules (2) We apply such architecture in the case of the visual-Sudoku task (given an image of a incomplete Sudoku board, the goal is to provide a complete symbolic solution) considerably improving the state-of-the-art (3) Finally, we show that when wrapping an existing state-of-the-art (replacing Neuro-Solver), our framework significantly improves its model performance.

The code is available at: `https://github.com/corneliocristina/NASR`.

## 1.1 RELATED WORKS

There has been a lot of recent research regarding the imposition of constrains in neural networks. This can be roughly divided in the following categories: **1) Modification of the Loss function:** Xu et al. (2018) add a component to the loss function quantifying the level of disagreement with the constraints; A similar idea can be found in the work of Xie et al. (2019) and Li et al. (2019); Wang & Pan (2020) instead exploit a parallel neuro-reasoning engine to produce the same output as the neural process and then add the distance between the two outcomes in the loss. **2) Adversarial training**: In the work of Ashok et al. (2021) they integrate a NN with a violation function generating new data; A similar idea can be found in the work of Minervini & Riedel (2018). **3) Adding an ad-hoc constraint output layer:** One example is adding a layer to the network which manipulates the output enforcing the constraints Giunchiglia & Lukasiewicz (2021); Ahmed et al. (2022) adds a compiled logic circuit layer to the network enforcing the constraints; Yang et al. (2020) and Manhaeve et al. (2018) instead create a parallel between the logic predicates with their neural version. **4) Logic relaxations:** Daniele & Serafini (2019) use a differentiable relaxation of logic to extend a NN with an output logic layer to increase the probability of compliant outcomes; Li & Srikumar (2019) have a similar approach but on the internal layers, augmenting the likelihood of a neuron when a rule is satisfied; Wang et al. (2019) introduce a differentiable MAXSAT solver that can be integrated into neural networks; Similar ideas using different types of logic relaxations can be found in the work of Gan et al. (2021); Sachan et al. (2018); Donadello & Serafini (2019) and Marra et al. (2019). **5) Neuro-symbolic integrations:** The alternating of purely symbolic components with neural ones can be found in the work of Agarwal et al. (2021) where the authors create an encoder/decoder with a standard reasoner in the middle. In another approach, Yang et al. (2020) combine answer set programs with the output of a neural network. Similar methods are the one of Tsamoura et al. (2021) and of Manhaeve et al. (2018). Other neuro-symbolic integration methods worth mentioning (that consider the specific show case of the Visual Sudoku task) are: Brouard et al. (2020) extract preferences from data and push it into Cost Function Networks; Mulamba et al. (2020) combine a neural perception module with a purely symbolic reasoner; and Yiwei et al. (2021) use curriculum-learning-with-restarts framework to boost the performance of Deep Reasoning Nets.

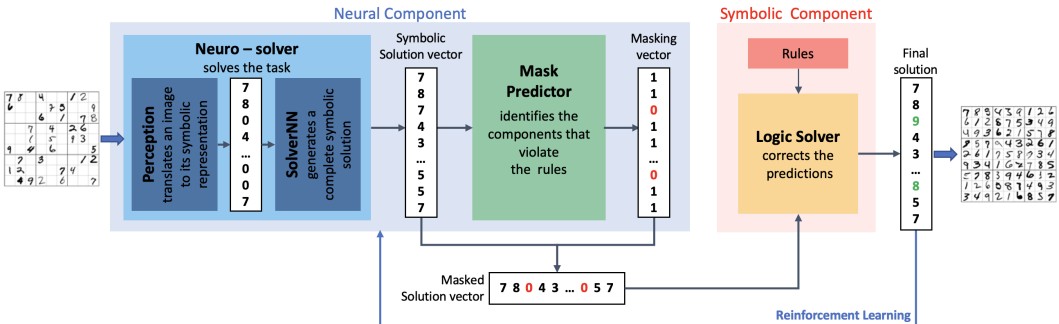

Figure 1: Pipeline for solving visual-Sudoku with our Neural Attention for Symbolic Reasoning.

## 2  METHOD

In this work we consider the type of tasks where multiple interpretable "facts" are predicted by a neural model on which the imposition of hard constraints is desirable. More formally, we consider a set of input data points ($x \in \mathcal{X}$) representing instances to solve (e.g. the picture of a partially filled Sudoku board), and, a set of multi-dimensional output data points ($y \in \mathcal{Y}$) that correspond to complete interpretable (symbolic) solutions (e.g. the symbolic representation of a completely filled Sudoku board). The collection of $N$ of these pairs of data points will form the *task dataset* $D = \{x^i, y^i\}_{i=1}^N$. Moreover, we require that the task (e.g. completing a partially filled Sudoku board) can be expressed (fully or partially) by a set of rules $\mathcal{R}$ in the form of domain-knowledge constraints (e.g. the rules of the Sudoku game).

The goal is to learn a function $f : \mathcal{X} \to \mathcal{Y}$ that associates a solution to a given input instance, *and which further satisfies the rules $\mathcal{R}$*. To solve this class of problems, we propose a neuro-symbolic pipeline that integrates 3 components, the *Neuro-Solver*, the *Mask-Predictor* and the *Symbolic-Solver* and that works as follows: An input instance is first processed by the Neuro-Solver that outputs an approximate solution. The solution is then analyzed by the Mask-Predictor that has the role of identifying the components of the Neuro-Solver predictions that do not satisfy the set of domain-knowledge constraints/rules $\mathcal{R}$. The masking output of the Mask-Predictor is then combined with the probability distribution predicted by the Neuro-Solver. This is done by deleting the wrong elements of the predictions, leaving the corresponding components "empty" (the component is filled by an additional class 0, indicating a masked element). This masked probability distribution is then fed to the Symbolic-Solver that fills the gaps with a feasible solution (satisfying the constraints/rules $\mathcal{R}$). In brief, the role of the Symbolic-Solver is to correct the Neuro-Solver prediction errors identified by the Mask-Predictor.

More formally: **1)** The *Neuro-Solver* is a function $ns(\cdot)$ that that maps an input $x \in \mathcal{X}$ (where $\mathcal{X}$ is the set of all possible inputs for the task under consideration) to a probability distribution over $\mathcal{Y}$ (where $\mathcal{Y}$ is the set of all the possible complete solutions); **2)** The *Mask-Predictor* is a function $mp(\cdot)$ that takes in input a probability distribution over $\mathcal{Y}$ and produce as output a probability distribution over $\mathcal{Z} = [0, 1]^k$ (where $k$ is the dimension of $y \in \mathcal{Y}$); **3)** The *Symbolic Solver* is a function $sb(\cdot)$ that maps $\mathcal{Y}'$ (where $\mathcal{Y}'$ is $\mathcal{Y}$ with an additional class 0, corresponding to a masked solution element) to a probability distribution over $\mathcal{Y}$.

The final hypothesis function $f_\theta$, mapping $\mathcal{X}$ to a probability distribution over $\mathcal{Y}$ and representing the neuro-symbolic pipeline approximating the target function $f(\cdot)$, is defined as:

$$f_\theta(x) = sb\big(\ ns(x) \odot \arg\max(mp(ns(x)))\big)\ ,\ \ \mathcal{R}\ \big) \tag{1}$$

where $\odot$ is the Hadamard (element wise) product and $\theta$ are the learnable parameters of $ns$ and $mp$.[1]

Figure 1 shows the pipeline architecture for the Visual Sudoku task.

**Example 1** *An example application is the visual Sudoku task (Figure 1). This task consists of providing a complete Sudoku board $y \in \mathcal{Y}$ corresponding to the solution of an incomplete input board*

---

[1]This assumes the solver expects 0 to indicate a symbol to fill. If the solver expects another symbol (or more generally), we can add an adapter function as $f_\theta(x) = sb\big(\ adapt(\ ns(x), \arg\max(mp(ns(x))))\ ,\ \mathcal{R}\ \big)$

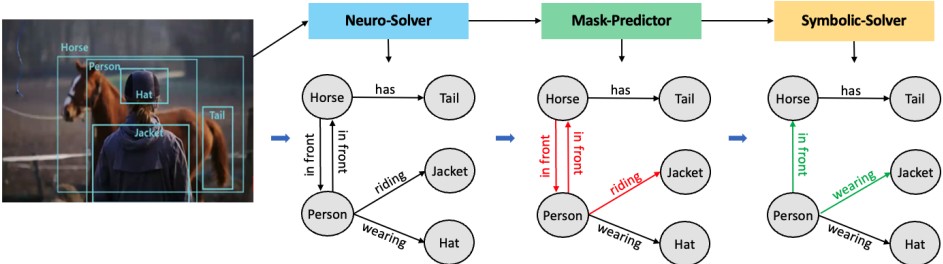

Figure 2: Motivating example: Predicate classification in scene graphs. The Neuro-Solver predicts a set of triples between the given objects. The Mask-Predictor identifies the components that potentially violate domain rules (e.g. the predicate *in front* is not symmetric and the object *jacket* is not in the range of the predicate *riding*). Finally, the Symbolic-Solver corrects the predictions.

*in the form of an image $x \in \mathcal{X}$. $\mathcal{X}$ is defined as $[0,1]^{252 \times 252}$ and corresponds to the set of the images of a Sudoku board (each Sudoku cell has dimension $28 \times 28$). $\mathcal{Y}$ is defined as $\{1, \ldots, 9\}^{81}$ and corresponds to the set of symbolic solutions where each cell is one of the possible 9 Sudoku digits. $\mathcal{Z}$ is defined as $\{0,1\}^{81}$. $\mathcal{Y}'$ is defined as $\{0, \ldots, 9\}^{81}$ and corresponds to the set of symbolic solutions with the 9 possible Sudoku digits and the digit 0 indicating empty cells. $\mathcal{R}$ contains the Sudoku rules: each cell needs to be filled with numbers in $\{1, \cdots, 9\}$, without repeating any numbers within the row, column or block. These can be formalized in different ways, depending on the Symbolic-Solver choice.*

**Example 2** *Another example application is the the Predicate Classification task (Figure 2). This task consists of predicting the right predicate between a set of objects (given in input in the form of labeled and localized bounding boxes) in an image. $\mathcal{X}$ corresponds to the input images with the set of labeled bounding boxes. This can be vectorized in different ways using appropriate embedding techniques (e.g. see Zellers et al. (2018)). $\mathcal{Y}$ is defined as $(\mathcal{B} \times \{1, \ldots, m\} \times \mathcal{B})^k$ and corresponds to the set of solutions. A solution is a set of $k$ triples with an object pair and a predicate between them (chosen within the set of $m$ possible predicates). $\mathcal{B}$ is the space of all possible labeled bounding boxes and is defined as $\mathbb{R}^4 \times \{1, \ldots, n\}$ where $n$ is the number of possible objects labels. $\mathcal{Z}$ is defined as $\{0,1\}^{k2}$. $\mathcal{Y}'$ is defined as $(\mathcal{B} \times \{0, \ldots, m\} \times \mathcal{B})^k$ and corresponds to the set of symbolic solutions $\mathcal{Y}$ augmented with the class 0 indicating the "empty" predicate (to be filled by the Symbolic-Solver). $\mathcal{R}$ is an ontology describing the set of object and predicates in the dataset. Some examples are type-rules constraining the domain and range of the predicates (e.g. the object cat is not in the domain of the predicate riding) or the symmetry/ reflexivity rules (e.g. the predicate in front is not symmetric).*

**Pipeline qualities and limitations.** This architecture is particularly useful for solving the tasks in which exhaustive (probabilistic) reasoning is not feasible (in most real-word scenarios standard reasoning techniques are not scalable) while purely neural architectures are not accurate enough (making mistakes that violates the domain-knowledge constraints/rules). By performing a quick approximate solution, and then focusing the attention of the expensive reasoning engine only on the parts that will benefit from reasoning, we achieve a favourable efficiency-accuracy trade-off.

By adding the symbolic component, we increase the performance at the cost of additional of running time, which will be higher compared to a purely neural model. We discuss this trade-off in more details in Section 4.2 Since the attention is done by a neural model (Mask-Predictor), there is no guarantee that all of the prediction mistakes will be found. However, some techniques can be used to minimize this risk (see Section 3.2).

Given the modularity of our NASR pipeline, it is possible to substitute each component with alternative ones. We will see an example of this in Section 4 where we integrate the SatNet method (Wang et al., 2019) in our pipeline and significantly improve its performance. For the same reason, our pipeline works independently of the input type, that can be imagery, text, symbolic, etc.

---

[2]The Hadamard product is intended between the predicate vector $p \in \{1, \ldots, m\}^k$ part of a solution $y \in (\mathcal{B} \times \{1, \ldots, m\} \times \mathcal{B})^k$ and the mask vector $m \in \{0,1\}^k$, since bounding boxes/labels are given in input.

## 2.1 Learning paradigm

Learning is done in two steps: the Neuro-Solver and the Mask-Predictor are first pre-trained individually in a supervised fashion and then integrated together and refined using reinforcement learning (RL). We refer to our complete pipeline as NASR, and disambiguate ablations where appropriate.

**Supervised Learning.** To train the Neuro-Solver we use the *task dataset* $D = \{x^i, y^i\}_{i=1}^N$ (also used for the whole pipeline). For training the Mask-Predictor we generate a synthetic dataset $D_{mp} = \{y_n^i, m^i\}_{i=1}^{N'}$ where: $y_n$ is a symbolic solution instance with the addition of noise that violates the domain-knowledge constraints; and $m$ is the corresponding masking solution. A masking vector $m$ has the same dimension of the input $y_n$ and has a 1 on the components of $y_n$ that do not violate the rules $\mathcal{R}$ and 0 for the components in which noise has been introduced.

The generation of $D_{mp}$ can be done in different ways depending on the type of data we are considering: 1) the input data $y_n$ can be either generated by perturbing the $y$ in $D$ or 2) it can be generated synthetically following a uniform distribution over the possible $y_i$ in $D$. In the former option, each data point $y_n \in D_{mp}$ has a corresponding data point $y \in D$ of which some components have been modified. The corresponding masking vector $m$ will have a 1 on the components of $y_n$ that has not been modified and 0 for the components in which noise has been introduced. In general, the latter option is not always possible: for example, in the case of the visual Sudoku task (see Example 1), this would require the ability to sample minimal[3] symbolic Sudoku boards uniformly at random, which is still a non-trivial open problem.

**Symbolic Solver.** The Symbolic-Solver will reason about the subset of outputs identified by the Mask-Predictor. The choice of the Symbolic Solver is strongly connected to the type of constraints/rules: For logic based constraints, classical symbolic reasoners can be used, such as Prolog engines (e.g. SWI-Prolog swi), probabilistic logic engines (e.g. ProbLog, Raedt et al.2007), python libraries that mimic symbolic reasoning engines (e.g. PySwip, Tekol & contributors2020), theorem provers (e.g. Vampire prover Riazanov & Voronkov2002), etc.; For arithmetic constraints, constraints-solvers can be used (e.g. ILP or MILP solvers), general mathematical tools (e.g. Mathematica mat) or ad-hoc brute force algorithms that exhaustively explore the symbolic solution search space. In this work we mostly consider logic rules and ontologies.

**Reinforcement Learning.** While the Neuro-Solver and Mask-Predictor can be trained independently with supervised learning, the use of reinforcement learning is necessary for end-to-end learning (since the Symbolic-Solver is not differentiable). End-to-end learning is important so that the neural components can adapt to the expected interventions of the Symbolic-Solver. In this work we use the REINFORCE algorithm Williams (1992), with its standard policy loss: $\mathcal{L}(x; \theta) = -r \log P_\theta(m|ns(x))$ where $r$ indicates the RL reward obtained when applying the Symbolic-Solver on the prediction $ns(x)$ masked by $m$ (for the Visual Sudoku case, see eq. 4). However, it is possible to use an alternative reinforcement learning algorithms without changing the overall pipeline.

## 3 Experiments Setup

### 3.1 Visual Sudoku Datasets

We considered a total of four Sudoku datasets, summarised in Table1. Three were drawn from an online repository online[4]: 1) *big_kaggle* contains 100'000 puzzles. It is a subset of a bigger dataset containing 1 million boards hosted on Kaggle[5] targeting ML models. These puzzles have an average of 33.82 hints per board (between 29 and 37); 2) *minimal_17* is a complete or nearly-complete dataset[6] of minimal Sudoku boards,with only 17-clues (17 is the minimum number of hints in a Sudoku board leading to a unique solution); and 3) *multiple_sol* containing 10'000 puzzles with two or more solutions. We converted these symbolic datasets into images using MNIST digits LeCun

---

[3]In minimal Sudoku boards, hints cannot be removed without losing uniqueness of the solution.

[4]https://github.com/t-dillon/tdoku

[5]https://www.kaggle.com/datasets/bryanpark/sudoku

[6]https://web.archive.org/web/20131019184812if_/http://school.maths.uwa.edu.au/~gordon/sudokumin.php

| dataset | # hints [min - max] | size | # of solutions | challenge |
|---|---|---|---|---|
| big_kaggle | 33.82 [29-37] | 100'000 | unique | scaling |
| minimal_17 | 17.00 [17-17] | 50'000 | unique | minimal number of hints |
| multiple_sol | 34.75 [34 -35] | 10'000 | multiple | multiple solutions |
| satnet_data | 36.22 [31-42] | 10'000 | unique | - |

Table 1: Sudoku datasets statistics.

& Cortes (2010). 4) Finally, we considered the dataset released with SatNet(Wang et al., 2019) (denoted as *satnet_data*). This dataset contains 10'000 Sudoku boards (both in image and symbolic form) with an average number of hints equal to 36.22 (minimum 31 and maximum 42).

We generated the $D_{mp}$ datasets for the Mask-Predictor by introducing noise in the task instances in $D$: for each board $y$ in the original dataset $D$ we generated a noisy board $y_n$ adding noise both in the the hint cells (swapping some of the hints digits) as well as in the solution cells (swapping some of solution cells). We perturbed these two sets separately. For smaller datasets (e.g. *satnet_data* and *multiple_sol*) we generated multiple masking boards per original board: for each board $y$ in the $D$ we generated $k = 10$ noisy board $y_n^1, y_n^2, \cdots, y_n^k$.

## 3.2 IMPLEMENTATION DETAILS AND COMPETITORS

The **Neuro-Solver** is the first component of the pipeline and has the role of producing a full symbolic solution from a input image board. It takes in input an image $x_i \in \mathcal{X}$ and output a probability distribution over the set of possible complete Sudoku solution boards $\mathcal{Y} = \{1, \ldots, 9\}^{81}$. In our implementation we divided this module in two sub-components: the **Perception** and the **SolverNN**.

The **Perception** model parses an input image of $252 \times 252$ into its symbolic representation of $9 \times 9 \times (9 + 1)$. In our implementation the Perception is a straightforward extension of the convolutional neural network (CNN) for single MNIST digit prediction LeCun et al. (1998) to multi-digit prediction by applying the CNN to each patch of $28 \times 28$. Formally, the Perception has an output layer of 81 nodes returning a 10-dimensional (the 9 input hints digits and the class 0 for the empty cells) posterior distribution over the digits $0 - 9$ for each input image $x_i \in \mathcal{X} = [0, 1]^{252 \times 252}$ of an incomplete Sudoku board. The output is then a probability distribution over $\mathcal{Y}' = \{0, \ldots, 9\}^{81}$. Each MNIST digit of the input image (corresponding to a single cell) is classified by the CNN independently[7]. The Perception model is trained to minimize the negative log likelihood loss (with log softmax as output layer), and is optimized via the ADADELTA optimizer.

The **SolverNN** model takes as input a probability distribution over $\mathcal{Y}' = \{0, \ldots, 9\}^{81}$ the set of partially filled Sudoku boards and outputs a probabilistic solution belonging to the space $\mathcal{Y} = \{1, \ldots, 9\}^{81}$. In our implementation **SolverNN** is a Transformer model with a linear layer mapping the input distribution (which is the output of the Perception) to 81 tokens/nodes of length 196, followed by 4 sequential self-attention blocks on top of these tokens; the input is positionally encoded to preserve the spatial information presented in the $9 \times 9$ Sudoku board (e.g the concepts of row, column, and $3 \times 3$ block). The output of the SolverNN is a probability distribution over the possible complete solutions $\mathcal{Y} = \{1, \ldots, 9\}^{81}$: 81 output nodes returning a 9-dimensional posterior distribution over the digits $1 - 9$. The SolverNN model is trained to minimize the Binary Cross Entropy with logits loss, and is optimized via Adam optimizer with weight decay.

The **Mask-Predictor** is a Transformer, with the same architecture and training loss as the SolverNN, with the only difference on the dimension of the output layer (1-dimensional). The Mask-Predictor takes in input a probability distribution over the set of complete solutions $\mathcal{Y} = \{1, \ldots, 9\}^{81}$, the output of the SolverNN model (81 nodes returning a 9-dimensional posterior distribution over the digits $1 - 9$, that can also be the one-hot representation of completely filled Sudoku boards when trained alone and not in the pipeline) and outputs a probability distribution over $\mathcal{Z} = \{0, 1\}^{81}$. For the Mask-Predictor, we add more weight to negative examples in the loss, since it is preferred to mask more cells, at the cost of masking correct ones, rather than non masking some of the errors in

---

[7]Note that the digits are treated independently just for simplicity and that the image can be processed in its entirety without significantly changes to the overall pipeline.

the SolverNN predictions. This is because, given that we are considering a non-probabilistic logic solver, a single error in the Symbolic-Solver input will make it fail.

The **Symbolic-Solver** takes in input a masked solution vector (the element-wise product of the output of the Neuro-Solver and the output of the Mask-Predictor) which is a partially filled 81-dimensional symbolic vector. The Symbolic-Solver will then attempt to solve it using the rules $\mathcal{R}$ providing a full symbolic solution vector as output. We considered two (non-probabilistic[8]) Symbolic-Solvers: 1) *PySwip*, a Python library to interface with SWI-Prolog, a reasoning engine for the Prolog programming language; 2) a brute force backtrack-based algorithm. Since we use a non-probabilistic Symbolic-reasoner, our hypothesis function corresponds to eq. 1 with an additional $argmax$ (since we cannot provide the full probability distribution over the classes, but only the most probable value for each cell):

$$f_\theta(x) = sb\big(\ \arg\max(ns(x)) \odot \arg\max(mp(ns(x)))\ ,\ \mathcal{R}\ \big)\ . \tag{2}$$

In the case in which an unsatisfiable neural prediction is given in input to the Symbolic-Solver, no solution is generated an thus the whole board is counted as wrong.

**Learning Algorithm (and RL).** As mentioned in Section 2 we first train the models in a fully supervised manner (details above) and then use reinforcement learning to refine the pipeline end-to-end. We define the RL scenario as follows: The input state $y_t$ corresponds to a solution board provided by the Neuro-Solver (Perception+SolverNN). The action space corresponds to the set of all possible complete masking boards configurations $\mathcal{Z} = \{0,1\}^{81}$. Each action $m$ coincides with the simultaneous execution of 81 independent sub-actions corresponding to the decision of masking or not a single cell in the solution board $y_t$. For each Sudoku board we sample an action $\tilde{m} \in \mathcal{Z}$ following the policy distribution. The final state $y_{t+1}$ corresponds to the solution board provided as output by the Symbolic-Solver (with input $\mathcal{R}$ and $y'_t = \arg\max(y_t) \odot \tilde{m}$). The action we perform corresponds to the application of the masking over the solution board provided by the Neuro-Solver followed by the deterministic application of the Symbolic-Solver to the masked solution board.

We use REINFORCE (Williams, 1992), with its standard policy loss. For each batch $B$, we have:

$$\mathcal{L}(B;\theta) = -\sum_{x\in B} r \log P_\theta(\tilde{m}|ns(x)) = -\sum_{x\in B}\Big(r\sum_{i=0}^{81}\log P_\theta(\tilde{m}_i|ns(x))\Big) \tag{3}$$

We use only positive rewards. Given an output board $b'$ and its ground truth board $b$, we consider two types of rewards with two different order of magnitude: the main reward, $r_e \in \{0,10\}$, when the entire board is correct and a marginal reward $r_c \in [0,1]$ for each correct cell $i$.

$$r = r_e + r_c = 10 \cdot \delta_{b',b} + \frac{1}{81}\sum_{i=0}^{81}\delta_{b'_i,b_i} \tag{4}$$

We normalize the rewards to improve training stability, since each board difficulty can vary.

**Baselines.** We compare NASR with different baselines. As *Symbolic Baseline* we considered the execution of the Symbolic-Solver directly from the output of the Perception module (after applying the $\arg\max$ operator). Another possibility would be adapting the *prediction-correction* part of the pipeline in the work of Giunchiglia et al. (2022a). Using a probabilistic reasoning engine and using the whole (or partial, e.g. the top $k$ candidates ) output distribution of the Perception module as input, is computationally unfeasible. We also compared with two state-of-the-art neuro-symbolic methods: *SatNet* Wang et al. (2019), a differentiable MAXSAT solver that can be integrated into neural networks; and *NeurASP* Yang et al. (2020), an extension of answer set programs (ASPs) that consider a neural network output as the probability distribution over atomic facts in ASPs.

## 4 RESULTS

The main results can be summarized as follows: 1) we outperform the baseline in most of the cases (and never perform worst); 2) we improve the performance of an existing method, by integrating it in our pipeline; 3) we are more efficient, compared to the other methods, in terms computational time vs. performance; and 4) our method is more robust to noise compared to the symbolic baseline.

---

[8]A probabilistic Symbolic-Solver is unfeasible in this scenario due to the combinatorial nature of the possible Sudoku boards.

|  | big_kaggle | minimal_17 | multiple_sol* | satnet_data |
|---|---|---|---|---|
| Symbolic Baseline | 74.56 | **87.70** | 63.50 | 63.20 |
| SatNet (Wang et al., 2019) | 63.44 | 0.00 | 0.00 | 60.10 |
| SatNet (Wang et al., 2019) + NASR | 69.05 | 0.02 | 24.20 | **81.40** |
| NeurASP† (Yang et al., 2020) | timeout | †**89.00** | timeout | timeout |
| Our NASR | **84.24** | 87.00 | **73.00** | **82.20** |

Table 2: Results for the visual Sudoku task: percentage of completely correct solution boards. The best results are in bold font. *The pipeline is underperforming on *multiple_sol* dataset likely due to the fact that each input board admits more than one solution but only one is provided at training. †tested only on 200 of the 5000 test images due to the long run-time (days).

|  | big_kaggle | | minimal_17 | | multiple_sol | | satnet_data | |
|---|---|---|---|---|---|---|---|---|
| Perception | 99.64 | 74.56 | 99.84 | 87.70 | 99.48 | 65.70 | 99.32 | 63.20 |
| SolverNN | 100-98.33 | 62.68 | 100-61.56 | 0.00 | 100-93.72 | 46.70 | 100-94.84 | 24.00 |
| Mask-Predictor | 99.92-99.71 | | 99.54-35.26 | | 99.02-76.12 | | 99.90-96.06 | |
| Neuro-Solver | | 47.03 | | 0.00 | | 28.80 | | 14.40 |
| NASR w/o RL | | 80.02 | | 1.59 | | 60.00 | | 76.40 |
| NASR with RL | | **84.24** | | **87.00** | | **73.00** | | **82.20** |
| NASR-Heur.Mask | | 73.11 | | 66.99 | | 55.60 | | 42.80 |

Table 3: Results of our pipeline on the visual Sudoku datasets, divided by the different modules. The metric used in the right column for each dataset correspond to the percentage of completely correct predicted boards. In addition to that we report (in the left column of each dataset): for the **Perception** the *number of correctly predicted cells on average*; for the **SolverNN** *preservation of input cells - correctness of solutions cells*; for the **Mask-Predictor** *true negative - true positive* (that corresponds to *correct solution cells that are not masked - error solution cells that are masked*).

## 4.1 OVERALL PERFORMANCE

In Table 2 we report the performance of our pipeline (with RL) on the different datasets compared with the different baselines. We can see that we outperform all the neuro-symbolic methods and in one instance match the Symbolic Baseline (which we outperform in all the other datasets). Note that symbolic baseline fails if even one digit is incorrectly recognised, and thus it is not noise robust. NASR can be integrated with SatNet (Wang et al., 2019) by replacing our Neuro-Solver with SatNet. The results (SatNet + NASR) show that the soft constraints enforced by SatNet can be improved, sometimes substantially by injection of hard constraints via NASR.

In Table 3 we report the performance of our pipeline analyzing each module separately. We can see that for more challenging datasets (e.g. *minimal_17* with only 17 hints minimal Sudoku boards and *multiple_sol* with multiple solutions) the performance of the Neuro-Solver is very low. In particular, for the *minimal_17* dataset this is even more evident, since the SolverNN cannot produce any completely correct solution board (only 60% of the solution cells is correct on average per board) and the Mask-Predictor identifies only 30% of the errors. RL is necessary to refine all the parts together.

Comparing Table 2 and Table 3 we observe that, as expected, the performance of the Symbolic Baseline corresponds to the performance (in terms of percentage of completely correct converted boards) of the Perception module. This is because, given the use of a non-probabilistic logic solver, a single perception error will make the Symbolic-Solver fail. The Symbolic baseline is thus not noise robust and finds datasets with more than the minimal number of hints harder to solve. Section A.2 systematically analyses the greater noise robustness of our framework to the symbolic baseline.

We performed an ablation study to verify the impact of the Mask-Predictor by substituting it with a simple heuristic. We masked Neuro-Solver outputs on the basis of confidence, with threshold set by grid search and Bayesian optimization (Akiba et al. (2019)). The results in Table 2 show that while this can perform well (e.g. in *big_kaggle*) it is significantly worse compared to our pipeline.

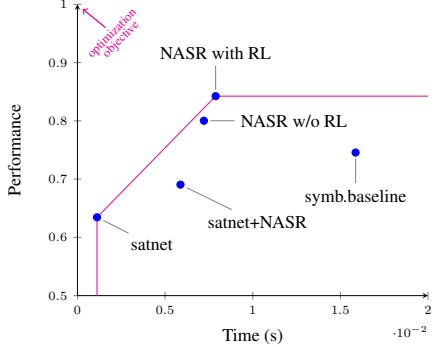 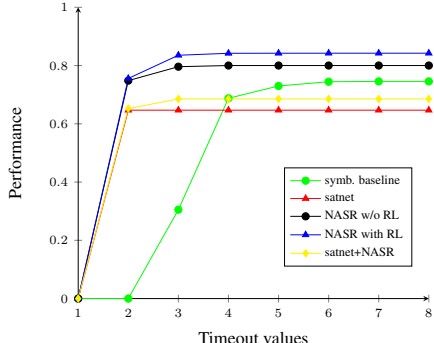

(a) Pareto front (purple line) maximizing the performance (percentage of completely correct boards) and minimizing the computational time. The optimization objective is located on the top left corner of the plot.

(b) Performance analysis for when limiting the computational time of the pipeline. The metric is the percentage of completely correct boards, while increasing the timeout limit.

Figure 3: Time efficiency analysis for *big_kaggle* dataset.

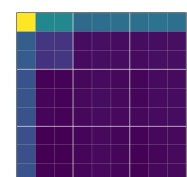 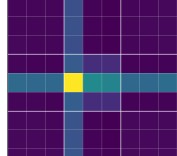 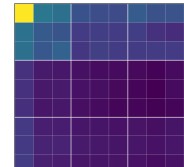 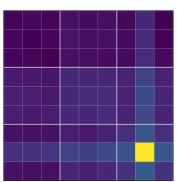

(a) SolverNN - Cell (1,1)    (b) SolverNN - Cell (5,4)    (c) Mask-Pred. - Cell (1,1)    (d) Mask-Pred. - Cell (8,8)

Figure 4: Attention maps for the SolverNN on *big_kaggle* and for the Mask-Predictor on *minimal_17*.

## 4.2 FURTHER ANALYSIS AND CONCLUSIONS

**Time efficiency.** Our system is faster than the symbolic baseline (Perception+Symbolic Solver). This is because our Symbolic Solver needs to fill less empty cells and thus its search space is reduced. In Figure 3 we analyse the efficiency in terms of trade-off between the performance (percentage of completely predicted Sudoku boards), and computational time for the *big_kaggle* dataset (see Supplementary Material for other datasets). Figure 3a shows the Pareto front considering the two optimization objectives of minimal computational time and maximal performance. Our method, is always on the Pareto front and usually is the closest to the optimization objective (top left corner of the plot). Figure 3b compares the performance of each system when limiting the computation time by different timeout values. We can see that with small timeout limits the neural models behave better compared to the Symbolic-Baseline which requires more time.

**Attention Maps.** To better understand the Neuro-Solver and Mask-Predictor networks, we analyse their attention maps for evidence of learning the rules of Sudoku. Figure 4 shows the average of all the attention layers for the SolverNN and for the Mask-Predictor on the *big_kaggle* and *minimal_17* dataset respectively. We can see that for each cell, both the SolverNN and the Mask-Predictor consider the information in the row, the column and the $3 \times 3$ block, which corresponds to the 3 Sudoku rules. The results are clearer for SolverNN on *big_kaggle* dataset compared to the Mask-Predictor on *minimal_17* dataset due to the dataset size.

**Preliminary Results on Scene Graph.** We performed a preliminary study for Predicate Classification (see Example 2) for the GQA dataset (Hudson & Manning (2019)) using a simple ontology containing only domain/range information for the predicates. The results are consistent with the Sudoku scenario (more information can be found in the Appendix A.8).

**Conclusions.** We presented a neuro-symbolic method that aims to efficiently satisfy domain-knowledge constraints at inference. This enables a favourable trade-off between accurate predictions, noise robustness, and computation cost. Our framework is generic and can be applied to different types of input (image, text, symbols, etc.) and types of constraints (logic, arithmetic, etc.).

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

# A APPENDIX

## A.1 TRAINING CONVERGENCE

In Figure 5 we provide the lerning curves for the reinforcement learning training on *minimal_17* dataset. In particular we can see in Figure 5b how the average reward on the training set improves in only few ($\sim 30$) iterations.

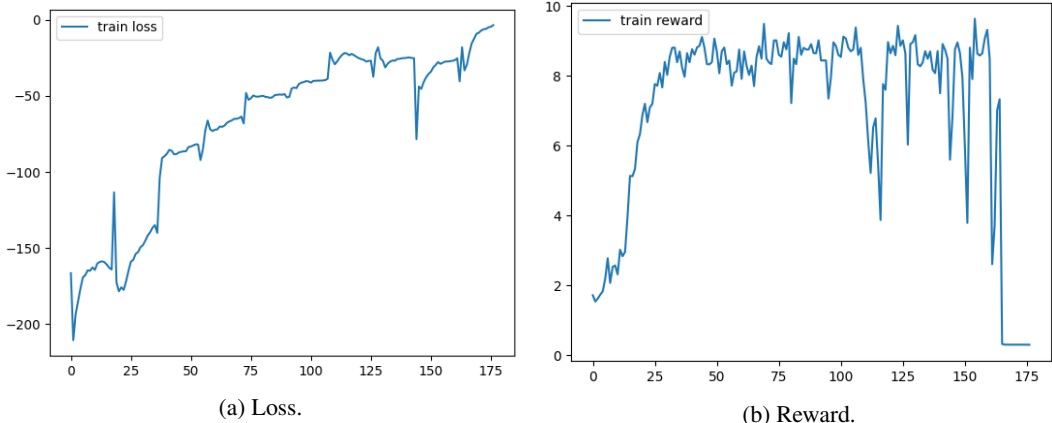

(a) Loss.                    (b) Reward.

Figure 5: Reinforcement learning curves for refining with Rl our pipeline on the *minimal_17* dataset.

Using bigger models can further improve the overall performance. For example, using a transformer with 12 attention modules (instead of 4) on the *big_kaggle* substantially improves the Neuro-Solver, achieving a boost of $\sim 30\%$ completely correct boards. This improvement is less evident considering the whole pipeline, gaining only a $\sim 5\%$ improvement compared to smaller models.

## A.2 ADDITIONAL ANALYSIS: NOISE ROBUSTNESS

Our pipeline is more robust to noise: in Figure 6 we can see the drop in performance of the baseline and our method adding two different type of noise in the input images at inference time. Figure 6b shows the results when adding Gaussian blur to the images, while Figure 6a when rotating the digits with a random angle in $[-45, 45]$. We can see that our pipeline has a smaller and slower drop in performance compared to the symbolic baseline[9]. These results are less evident in the refined pipeline using RL. We can see that with a high amount of noise the performance gap with the Symbolic-Solver is smaller (e.g. with a rotation between $[-40, 40]$ degrees, our pipeline refined with RL solves $20.3\%$ more boards compared to the Symbolic-Solver), while for a medium amount of noise the gap with the Symbolic-Solver is higher (e.g. with a rotation between $[-25, 25]$ degrees our pipeline refined with RL solves $32.2\%$ more boards compared to the Symbolic-Solver). The difference between the pipeline with or without RL is probably due to the fact that, in the latter, the Mask-Predictor is trained on simulated noise while in the former the Mask-Predictor is refined on the actual noise produced by the Neuro-Solver. Thus, when adding a new and unseen type of noise the latter is expected to behave better.

## A.3 ADDITIONAL RESULTS: EFFICIENCY.

As mentioned in Section 4.2, we can see that we obtain similar efficiency results for the remaining datasets: *minimal_17* (Figure 7); *multiple_sol* (Figure 8); and *satnet_data* (Figure 9).

---

[9]Note that these experiments are done training the Mask-Predictor with an expected amount of noise between 0% and 10%. The Mask-Predictor can be retrained and optimized for the specific amount of noise expected in input, leading to even better performance.

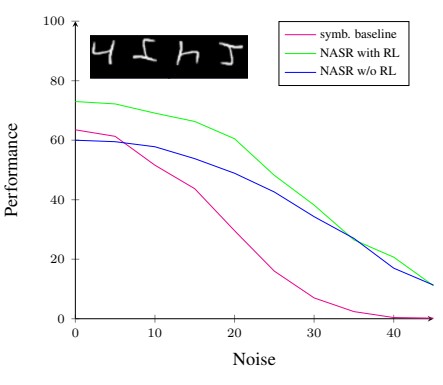

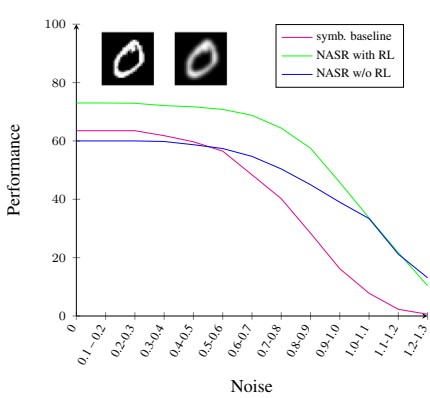

(a) Noise at test time: digit rotation.

(b) Noise at test time: Gaussian blur.

Figure 6: Performance of the Symbolic Baseline compared to our two pipelines (with or without RL) when adding noise on the dataset *multiple_sol* at test time. Similar results hold for the other datasets.

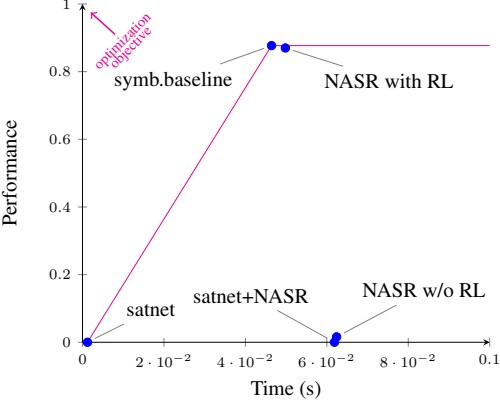

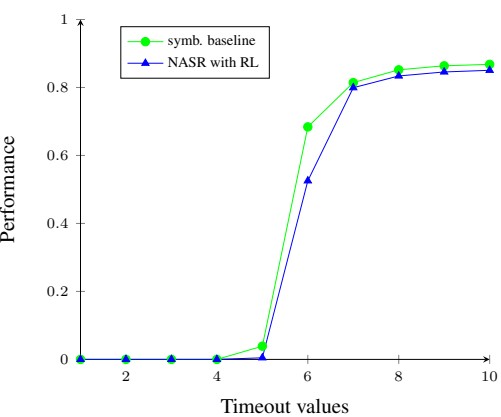

(a) Pareto front (purple line) for the *minimal_17* dataset maximizing the performance (percentage of completely correct boards) and minimizing the computational time. The optimization objective is located on the top left corner of the plot.

(b) Performance analysis for the *minimal_17* dataset when limiting the computational time of the pipeline. The metric is the percentage of completely correct boards, while increasing the timeout limit.

Figure 7: Time efficiency plots *minimal_17* dataset.

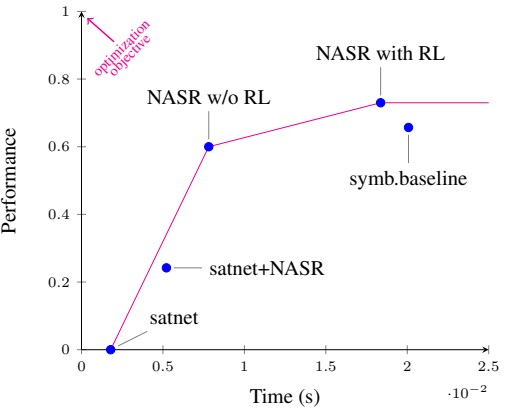 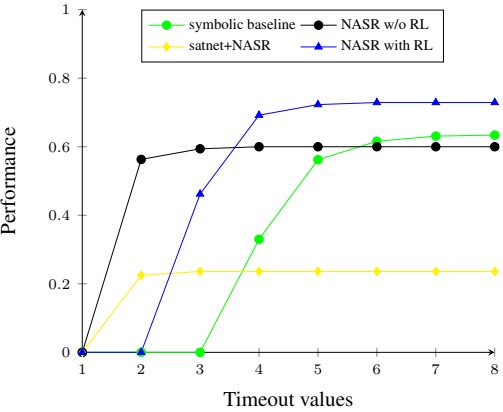

(a) Pareto front (purple line) for the *multiple_sol* dataset maximizing the performance (percentage of completely correct boards) and minimizing the computational time. The optimization objective is located on the top left corner of the plot.

(b) Performance analysis for the *multiple_sol* dataset when limiting the computational time of the pipeline. The metric is the percentage of completely correct boards, while increasing the timeout limit.

Figure 8: Time efficiency plots *multiple_sol* dataset.

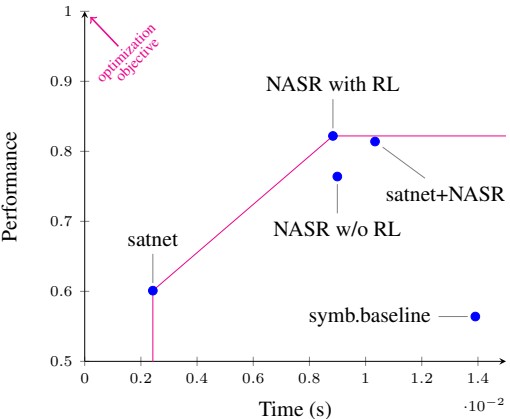 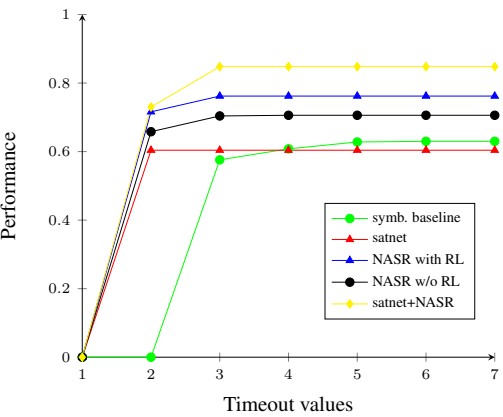

(a) Pareto front (purple line) for the *satnet_data* dataset maximizing the performance (percentage of completely correct boards) and minimizing the computational time. The optimization objective is located on the top left corner of the plot.

(b) Performance analysis for the *satnet_data* dataset when limiting the computational time of the pipeline. The metric is the percentage of completely correct boards, while increasing the timeout limit.

Figure 9: Time efficiency plots *satnet_data* dataset.

## A.4 ADDITIONAL RESULTS: ATTENTION MAPS.

Additional plots for the attention maps considering more cells are shown in Figure 10.

We remark that the attention maps depend on the dataset. In particular, if a dataset contains some bias (e.g. *multiple_sol*) from how it has been generated, the networks will try to exploit the biases (e.g. learning a relaxed version of the rules $\mathcal{R}$ or some simpler heuristic) instead of learning the proper Sudoku rules.

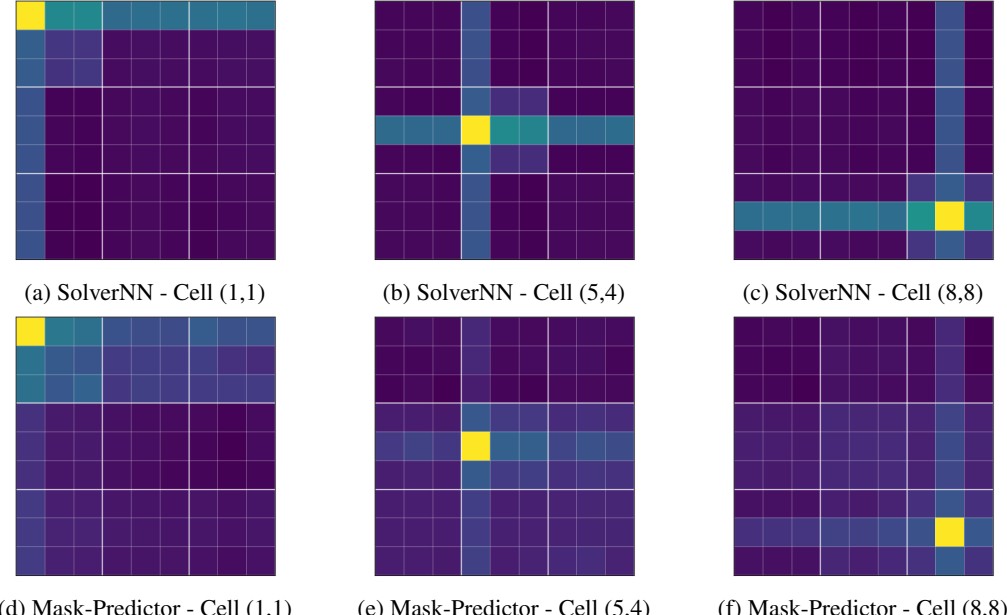

(a) SolverNN - Cell (1,1)          (b) SolverNN - Cell (5,4)          (c) SolverNN - Cell (8,8)

(d) Mask-Predictor - Cell (1,1)    (e) Mask-Predictor - Cell (5,4)    (f) Mask-Predictor - Cell (8,8)

Figure 10: Attention maps for the SolverNN on *big_kaggle* and for the Mask-Predictor on *minimal_17*.

Figure 11: Pipeline example for the visual Sudoku task. The Neuro-Solver outputs a complete board. The Mask-Predictor identifies the cells that violate the Sudoku rules (in the first block there are two cells with value 7). Finally, the Symbolic-Solver corrects the predictions.

### A.5 SUDOKU RULES FORMULATION FOR PYSWIP

In what follows we provide the rules used in *pyswip* (Tekol & contributors, 2020) for finding a symbolic solution for a partially filled Sudoku board[10]. The masked cells are provided as a "_" symbol. The query corresponds to the example shown in Figure 1 and Figure 11

```
% Sudoku rules

:- use_module(library(clpfd)).

sudoku(Rows) :-
      length(Rows, 9), maplist(same_length(Rows), Rows),
      append(Rows, Vs), Vs ins 1..9,
      maplist(all_distinct, Rows),
      transpose(Rows, Columns),
      maplist(all_distinct, Columns),
      Rows = [As,Bs,Cs,Ds,Es,Fs,Gs,Hs,Is],
      blocks(As, Bs, Cs),
      blocks(Ds, Es, Fs),
      blocks(Gs, Hs, Is),
      label(Vs). % default strategy min (follows order in Vs = 1..9)

blocks([], [], []).
blocks([N1,N2,N3|Ns1], [N4,N5,N6|Ns2], [N7,N8,N9|Ns3]) :-
      all_distinct([N1,N2,N3,N4,N5,N6,N7,N8,N9]),
      blocks(Ns1, Ns2, Ns3).

% query
Rows = [[7,8,_,4,3,9,1,2,6],
        [6,1,2,8,7,5,3,4,9],
        [4,9,3,6,2,1,5,_,8],
        [8,5,7,9,4,3,2,6,1],
        [2,_,1,7,5,8,9,3,4],
        [9,3,4,1,6,2,7,8,5],
        [5,7,8,3,_,4,6,1,2],
        [1,2,6,5,8,7,4,9,3],
        [3,4,9,2,1,6,_,5,7]],
      sudoku(Rows).
```

---

[10]This is a standard implementation of Sudoku rules, and it is a modified version of the one at `https://swish.swi-prolog.org/pldoc/man?section=clpfd-sudoku`

## A.6 Training Times

In Table 4 we report the training times for the different methods. The training time of the Symbolic Baseline corresponds to the training time of the Perception, since it needs the Perception output as input of the Symbolic-Solver. The training time for NASR is the sum of perception, SolverNN, Mask-Predictor, and optionally RL. The first three of these can trivially be parallelised. We trained the transformer models for 200 epochs, while the Perception models for 100 epochs. RL refinement is trained for 20 epochs. For SatNet and NeurASP we used the default configuration (100 epochs for both models). NeurASP-Infer uses our pretrained Perception to convert the image to symbolic form, while NeurASP uses their own CNN.

|  | big_kaggle | minimal_17 | multiple_sol | satnet_data |
|---|---|---|---|---|
| Perception | 108.84 | 57.71 | 11.68 | 14.14 |
| SolverNN | 85.58 | 41.97 | 9.01 | 26.51 |
| Mask-Predictor | 81.43 | 40.15 | 8.87 | 128.98 |
| RL | 121.07 | 241.31 | 24.86 | 16.44 |
| NASR w/o RL | 275.85 (108.84) | 139.83 ( 57.71) | 29.56 (11.68) | 169.63 (128.98) |
| NASR with RL | 396.92 (229.91) | 381.14 (299.02) | 54.42 (36.54) | 186.07 (145.42) |
| Symb. Baseline | 108.84 | 57.71 | 11.68 | 14.14 |
| SatNet | 1151.25 | 582.13 | 41.66 | 108.51 |
| NaurASP-Infer | 108.84 | 57.71 | 11.68 | 14.14 |
| NeurASP | 94.34 | 91.53 | 99.96 | $\sim 90$ |

Table 4: Training times (minutes) of the different systems for the visual Sudoku task. For NASR we report the training time when training the different models sequentially and in brackets the training time when training the Perception, SolverNN and Mask-Predictor modules in parallel.

## A.7 Analysis on the Different Roles of NASR Modules in Error Correction

In Table 5 we report the results of two experiments that help to understand how the different components of NASR interact with each other for the Visual Sudoku task.

In the first experiment we look at the input (hints) cells that have been incorrectly predicted by the Perception but have been corrected by NASR (2nd and 3rd column). The goal of this experiment is to see the proportion, of this type of errors, that are corrected directly by the SolverNN or thanks to the Mask-Predictor. We can observe that the Perception errors are almost all corrected by the SolverNN, while only a small portion are remain to be identified by the Mask-Predictor.

The second experiment (4th and 5th column) aims to understand if the Mask-Predictor distinguishes between the errors in the input cells (the original hints) and the errors in the solutions cells. We can see that the percentage of errors identified in the hint cells (4th column) and in the solution cells (5th column) has no distinctive pattern. We can conclude that the Mask-Predictor does not systematically prefer either of the two. It corrects both types of errors.

|  | Perception errors corrected by NASR w/o RL | | SolverNN errors identified by | |
|---|---|---|---|---|
|  | identified by | identified by | Mask-Predictor | |
|  | SolverNN | Mask-Predictor | hint cells | solutions cells |
| big_kaggle | 85.90 | 14.10 | 53.17 | 77.42 |
| minimal_17 | 95.90 | 4.10 | 8.21 | 32.24 |
| multiple_sol | 87.21 | 12.79 | 42.09 | 24.73 |
| satnet_data | 85.43 | 14.57 | 44.76 | 71.16 |

Table 5: Statistics on the interplay of the different component of the NASR pipeline in the case of Visual Sudoku task in regard to error correction. All the results corresponds to percentages. The reported results refer to NASR without RL. Similar results hold for NASR with RL.

## A.8 Preliminary Results on SceneGraph

We evaluate NASR w/o RL on the task of predicate classification (PredCls). Recall that the task consists in providing the right predicate label given the ground truth object labels and bounding boxes.

### A.8.1 Dataset and Baseline

We consider the GQA dataset introduced by Hudson & Manning (2019). GQA is a more balanced split of Visual Genome (VG) dataset (Krishna et al., 2016) with cleaner, larger, and more dense scene graphs and with a larger and more balanced variety of objects and predicates (for more details see Hudson & Manning (2019)). It contains 22M pairs of question/answer for real-world visual reasoning and the corresponding scene graphs. We are interested in the latter.

In the GQA dataset there are 311 predicates and 1705 classes for the subject/object element of a triple. Statistics of the train/val/test split that we used are provided in table 6. The *ZeroShot* versions of the validation and test set contains only triples that has not been observed during training.

|  | # images | # triples | # unique triples |
|---|---|---|---|
| train | 66078 | 3455468 | 470129 |
| val (all - shots) | 4903 | 306494 | 89411 |
| test (all - shots) | 10055 | 530326 | 147534 |
| val (zero - shots) | 3025 | 23371 | 18638 |
| test (zero - shots) | 6418 | 45135 | 37116 |

Table 6: GQA dataset statistics.

As baseline, we consider the model developed by Knyazev et al. (2020) which is a modified version (that exploits a density-normalized edge loss) of the Message Passing (MP) architecture introduced by Xu et al. (2017). We adopted the data split and evaluation metrics used in their work. For our experiments, we trained their models on the GQA dataset (the code is available at `https://github.com/bknyaz/sgg`).

### A.8.2 Implementation Details

The input and output spaces of the pipeline modules follow what was introduced in Example 2.

As **Neuro-Solver** we chose the architecture introduced by Knyazev et al. (2020) mentioned above. It takes in input an image with the corresponding labels and bounding boxes of the objects and outputs a probability distribution over the predicates for each identified triples. For more details about this architecture see the work of Knyazev et al. (2020).

The **Mask-Predictor** is a (4-layer) MLP model that embeds predicate classes and object/subject classes differently. It takes in input a probability distribution over the set of predicates (311-dimensional) and the one-hot encoding (1705-dimensional) of subject and object for each triple. The output of the Mask-Predictor is a probability distribution over $\mathcal{Z} = \{0, 1\}^k$. The number of triples $k$ can vary for every image (they are considered as independent from each other since we are considering a simple domain-range ontology).

The Mask predictor is trained on a synthetic dataset generated from the domain-range ontology (an ontology containing all the possible domain and range classes for each predicate). We sampled an equal number of triples for each of the following 4 scenarios: triples that do not violate the ontology, triples with both unfeasible domain and unfeasible range, triples with only unfeasible domain and triples with only unfeasible range.

The **Symbolic-Solver** is a simple solver, implemented in python, that takes in input a masked solution and finds the next most probable predicate, after excluding the not feasible ones, for a given object/subject pair.

### A.8.3 RESULTS

The results considering a simple domain-range ontology are provided in Table 7, using the standard image-level *Recall* metric and Table 8, using the image-level *mRecall* metric, introduced by Knyazev et al. (2020). *Recall* consider only the top-1 prediction (predicate) between each pairs of objects, while *mRecall* allows to rank multiple predictions (predicates) between each pairs of objects.

We report the results for the Baseline (the work of Knyazev et al. (2020)) and the improvement over the Baseline of by the Probabilistic Symbolic Baseline (PSB). PSB consists of running the probabilistic symbolic solver directly on the output of the Baseline model. This is computationally very expensive, especially if we consider a sightly more dense ontology. With a more complex ontology this would became computationally intractable. The improvement given by the PSB in the case of PredCl is an upper-bound (Max-improvement) for the performance of NASR. The results of NASR w/o RL are reported as percentage of error correction achieved when compared to the PSB upper bound.

The results show that NASR achieves good performance, and is able to recover the majority of the recoverable errors given the simple domain-range ontology used. This leads, for example, to a improvement between 1% and 2% for the zero-shots predictions. Since we are considering a very simple ontology, the improvement is not as noticeable as in the Visual Sudoku case. However when using a more complex ontology, we expect this difference to become more pronounced.

| | R@20 | R@50 | R@100 | R@200 | R@300 |
|---|---|---|---|---|---|
| **All - shots** | | | | | |
| Baseline Knyazev et al. (2020) | 29.22 | 42.35 | 48.48 | 50.75 | 51.11 |
| Max-improvement (PSB) | 0.12 | 0.23 | 0.32 | 0.35 | 0.36 |
| % improvement of NASR w/o RL | 99.71 | 99.58 | 99.69 | 99.64 | 99.64 |
| **Zero - shots** | | | | | |
| Baseline Knyazev et al. (2020) | 16.62 | 27.65 | 34.10 | 37.41 | 38.11 |
| Max-improvement (PSB) | 0.91 | 1.43 | 1.93 | 2.18 | 2.33 |
| % improvement of NASR w/o RL | 100.00 | 100.00 | 100.00 | 100.00 | 100.00 |

Table 7: Results on the PredCl task for the GQA dataset. We consider the Recall metric used both for VG and the GQA datasets (see Hudson & Manning (2019) for more details). NASR results are given as percentage of the max achievable improvement under the given ontology, defined by PSB.

| | mR@20 | mR@50 | mR@100 | mR@200 | mR@300 |
|---|---|---|---|---|---|
| **All - shots** | | | | | |
| Baseline Knyazev et al. (2020) | 36.24 | 60.84 | 79.08 | 91.12 | 94.76 |
| Max-improvement (PSB) | 0.18 | 0.33 | 0.50 | 0.51 | 0.56 |
| % improvement of NASR w/o RL | 98.09 | 91.20 | 91.16 | 90.25 | 87.90 |
| **Zero - shots** | | | | | |
| Baseline Knyazev et al. (2020) | 19.16 | 37.46 | 54.64 | 69.72 | 76.12 |
| Max-improvement (PSB) | 0.08 | 0.46 | 0.89 | 1.37 | 1.93 |
| % improvement of NASR w/o RL | 100.00 | 86.77 | 93.05 | 88.99 | 86.00 |

Table 8: Results on the PredCl task for the GQA dataset. We consider the new mRecall metric introduced for the GQA dataset by Knyazev et al. (2020). NASR results are given as percentage of the max achievable improvement under the given ontology, defined by PSB.

