# OpenReview forum: "Learning where and when to reason in neuro-symbolic inference"
_ICLR.cc/2023/Conference — ICLR 2023 notable top 5%_

### Official Review · Reviewer_ndKP · 2022-10-21

**Confidence:** 5
**Correctness:** 3
**Technical Novelty And Significance:** 3
**Empirical Novelty And Significance:** 3
**Recommendation:** 8

**Clarity, Quality, Novelty And Reproducibility:**

**Quality:**

The paper presents a good idea but the clarity of the paper and the experimental results can be improved. If that is done the quality of the paper can be greatly improved.
The are also some mistakes/things I struggle to understand:
1. At page 3 you write that $\mathcal{Z} = [0,1]^k$ where $k$ is the size of the input. Shouldn't it be the size of the output (i.e., $|\mathcal{Y}|$)?
2. The equation at page 3 and 6 are different. One is the Hadamard product between $ns()$ and argmax() while the second is the product of two argmax. Is there a reason behind this difference?
3. Also, what does it mean to do the argmax between two argmax?
4. It is not clear to me what would happen if the output of the neuro-solver would satisfy the constraints. In that case, can the mask-predictor change the prediction anyway or is there a check done that the masking cannot be done in that case?
5. In general, when $D_{mp}$ is created shouldn't there a check that the noise added actually create assignments that violate the constraints? E.g., suppose that I have the constraint $A \to B$ and that the correct assignment is $A=1, B=1$, I can add noise to this and generate $A=0, B=1$. This instance still satisfies the constraints, and thus it would not be helpful in training the mask predictor.
6. In the generation of the sudoku dataset for the mask predictor you write "In general, the latter option is not always possible: for example, in the case of the visual Sudoku task (see Example 1), this would require the ability to sample minimal symbolic Sudoku boards uniformly at random, which is still a non-trivial open problem.". Why would you need to sample a minimal symbolic Sudoku? Doesn't the mask-predictor anyway take as input the full output (i.e., the sudoku completely filled in)?
7. What is $\delta$ in the reward equation?
8. Can you add in the appendix how you encode the sudoku rules? (i.e., the full list and which fragment of logic has been used). Also how do you encode the rules that only the masked facts should be changed?
9. Would it be possible to also have the experiment on the scene graphs? (This wouldn't dramatically changed my assessment, however it would make the paper much more complete)
10. The authors have only proposed as baseline to add a solver on top of the perception module. However it is not the only possible way. In [1], for example, the authors show how it is possible to correct a complete prediction so that the constraints are guaranteed to be satisfied. Would it be possible to add such baseline?


**Clarity:**

The paper needs some rewriting. Here there are some suggestions:
- In the introduction itself, the authors should clearly state what is:
  1. The expressivity of the constraints admitted by the framework. It seems to be it is propositional logic, as the pipeline requires to have one element in the prediction of the neuro-solver for each atom (this implies that even if you admit predicates, the space of your variables is still finite, thus the predicates are just syntactic sugar)
  2. The exact machine learning task on which the authors are trying to impose the constraints (e.g., supervised, semi-supervised etc.). It seems to be the fully-supervised setting. An in particular a case of a multilalbel classification problem, where each output must be in $[0,1]$ and there can be multiple labels associated to each prediction.
- Figure 1 and example 2 are misleading. The authors implement and test only for the sudoku task, and they do not go into the details on how the architecture would look like for the problem of object detection with constraints. Thus they should be removed.
- The paper states that as long as there are no false negatives the constraints are satisfied. This I think would apply only in the case of object detection (while in the case of the sudoku, everything should work even with false negatives. Is that correct?)
- Figure 1 and 2 are misplaced. They should be moved earlier in the paper.
- I personally would change the "Neuro-Solver" to simply neural-network. This would make the pipeline much clearer. (In the end these will literally be one or more neural networks).
- You introduce the "Perception" module only in the "Experimental Setup" section, which might be confusing for the reader. I would again just talk about "neural network" and then say that for the particular sudoku task you have one that takes care of the perception and one to fill in the sudoku.

**Novelty:**

The paper presents a novel idea which has the potential to have an impact on the community. Some of citations are missing, see the list of references below.

**Reproducibility:**

Will the authors make the code publicly available?

**References:**
[1] Eleonora Giunchiglia, Mihaela Catalina Stoian, Salman Khan, Fabio Cuzzolin, Thomas Lukasiewicz. ROAD-R: The Autonomous Driving Dataset with Logical Requirements. Machine Learning, 2022.

[2] Nicholas Hoernle, Rafael-Michael Karampatsis, Vaishak Belle, and Kobi Gal. MultiplexNet: Towards fully satisfied logical constraints in neural networks. In Proc. of AAAI, 2022.

[3] Marc Fischer, Mislav Balunovic, Dana Drachsler-Cohen, Timon Gehr, Ce Zhang, and Martin Vechev. DL2: Training and querying neural networks with logic. In Proc. of ICML, 2019.

[4] Alessandro Daniele, Emile van Krieken, Luciano Serafini, Frank van Harmelen. Refining neural network predictions using background knowledge. Machine Learning, 2022.

[5] Paolo Dragone, Stefano Teso, and Andrea Passerini. Neuro-symbolic constraint programming for structured prediction. In Proc. of IJCLR-NeSy, 2021.

*Please update the paper accordingly (the current score has been given according to the potential of the paper)*
Note: the new experimental results are **not** a necessary addition.

**Strength And Weaknesses:**

**Strengths:**
- The paper presents the novel idea of selecting which facts need to be changed in order to make the prediction complaint with the constraints. This effectively reduces the search space for the solver, and it might be a useful intuition for future works.
- The presented method is intuitively easy to understand.

**Weaknesses:**
- The paper needs significant rewriting. It lacks a lot of details, and while intuitively is easy to follow, when it comes to the details it becomes a very fuzzy. This is the main problem I have with the paper, which I though think can be solved with some effort. See comments on Clarity below for more details on how to address the issue.
- The experimental analysis has been done on a single task which limits the understanding on how this model would perform in real-world datasets.




**Summary Of The Paper:**

The paper proposes a new model to enforce hard logical constraints on neural networks. The model proposed, called NASR, has three main components:

1) A neuro-solver: which is made of one (or more) neural networks. It takes as input the datapoints in the dataset and returns an initial prediction (which might contain violations of the constraints),

2) A Mask-predictor: which is a neural-based model that takes as input the initial prediction made by the neuro-solver at returns a mask over such output. The $i$th element of the mask is equal to 1 if the $i$th element needs to be changed in order to satisfy the constraints, and 0 otherwise.

3) A Symbolic Solver: which (as the name says) is a symbolic solver that will correct those elements of the prediction (or facts, as they are called in the paper) which have been identified by the mask-predictior to make the final output compliant with the constraints.

**Summary Of The Review:**

TL;DR: paper could improve a lot if some parts were rewritten and a lot of details clarified.

---

> ### Author Response · Authors · 2022-11-18
> **Rebuttal - part 1**
>
> **C: The paper needs significant rewriting. It lacks a lot of details, and while intuitively is easy to follow, when it comes to the details it becomes very fuzzy. This is the main problem I have with the paper, which I think can be solved with some effort.**
>
> Thanks. We have attempted to address all your specific comments.
>
> **Q(1): At page 3 you write that $Z=[0,1]^k$ where $k$ is the size of the input. Shouldn't it be the size of the output (i.e., $|Y|$)?**
>
> Thanks for noticing this typo, we fixed it.
>
> **Q(2): The equations on page 3 and 6 are different. One is the Hadamard product between ns and argmax while the second is the product of two argmax. Is there a reason behind this difference?**
>
> Yes, this is because in our implementation we used a non-probabilistic symbolic solver. For this reason, we cannot provide the full probability distribution over the output but just the most likely value for each cell. This is the reason for the additional argmax. We added this comment to the paper to make it more clear.
>
> **Q(3): Also, what does it mean to do the argmax between two argmax?**
>
> We do not apply an argmax between two argmax. We run the symbolic solver over the Hadamart product of the two argmax. In this setting, the Hadamart product between the two argmax is the element wise product of two vectors of length 81 (the neural solution and the mask vector).
>
> **Q(4): It is not clear to me what would happen if the output of the neuro-solver would satisfy the constraints. In that case, can the mask-predictor change the prediction anyway or is there a check done that the masking cannot be done in that case?**
>
> If the neuro-solver already satisfies all the constraints, the mask predictor should ideally output all $1$s. In practice it might output a small number of $0$s, which are then re-computed by the symbolic solver. (Recall that the mask predictor needs to be conservative to maintain a low false negative rate, which would constitute an unrecoverable error.)
>
> Currently there is no intermediate consistency check to see if the neuro-solver is completely correct and potentially bypass the masking if so. This could trivially be added, but would not substantially change the result.
>
> **Q(5): In general, when $D_{mp}$ is created shouldn't there be a check that the noise added actually creates assignments that violate the constraints? E.g., suppose that I have the constraint $A→B$ and that the correct assignment is $A=1,B=1$, I can add noise to this and generate $A=0,B=1$. This instance still satisfies the constraints, and thus it would not be helpful in training the mask predictor.**
>
> As mentioned in the paper, the process to generate $D_{mp}$ highly depends on the type of data we are considering. In some cases, like the one you mention, it would be necessary to have a slightly more sophisticated process to create the data. In the case of Sudoku, we consider, as noise, permutations of a subset of cells. This will ‘always’ lead to an infeasible solution: having a permutation on the randomly selected subset of cells that exactly corresponds to one of the sudoku symmetries on this particular group of cells is extremely unlikely (and thus will not influence the training).
>
> **Q(6): In the generation of the sudoku dataset for the mask predictor you write "In general, the latter option is not always possible: for example, in the case of the visual Sudoku task (see Example 1), this would require the ability to sample minimal symbolic Sudoku boards uniformly at random, which is still a non-trivial open problem.". Why would you need to sample a minimal symbolic Sudoku? Doesn't the mask-predictor anyway take as input the full output (i.e., the sudoku completely filled in)?**
>
> When we generate $D_{mp}$ synthetically, we want to recreate the original dataset distribution in the best way possible. In the dataset minimal_17, only minimal symbolic Sudoku are present. When we introduce errors we might want to introduce it differently for the perception part (the hints) and the solution part (filled by the neuro solver). Indeed, in our experiments we add different amounts of noise for these two components when we generate $D_{mp}$: the perception is very reliable in most cases, thus we do not want to introduce much error in that part; while we want to add more error in the part that was filled by the neuro-solver. Hence the necessity to have only 17-hints boards as part of the $D_{mp}$ generation process.
>
> In general (considering other types of data/dataset other than visual-Sudoku), an uniform sampling for a synthetic generation is not always possible.
>
> **Q(7): What is $\delta$ in the reward equation?**
>
> The $\delta$ in the reward equation is the Kronecker delta function:  $\delta_{i,j}=1$ if $i=j$, $0$ otherwise.

---

> ### Author Response · Authors · 2022-11-18
> **Rebuttal - part 2**
>
> **Q(8): Can you add in the appendix how you encode the sudoku rules? (i.e., the full list and which fragment of logic has been used). Also how do you encode the rules that only the masked facts should be changed?**
>
> As mentioned in the paper, we used two different ways to solve a sudoku board (depending on the dataset). For some dataset a simple brute-force backtracking algorithm is faster, while for others prolog is faster. We added the Sudoku rules we used in pyswip in Appendix A5.
>
> To obtain a solution with a masked prediction, we input a partially filled sudoku board and let the solver (backtrack or Prolog) fill the empty cells. The partially filled sudoku board corresponds to the neuro-solver solution where we deleted some cells: the empty cells are the ones predicted as $0$s by the mask-predictor.
>
> **Q(9): experiment on the scene graphs. The experimental analysis has been done on a single task which limits the understanding on how this model would perform in real-world datasets.**
>
> We added in the Appendix (see section A.8) some preliminary results on the Scene Graph domain (PredCl), as described in Example 2 and Figure 1. Existing publicly available ontologies are very simple, so the maximum achievable improvement with, even by exhaustive reasoning, is quite small. However, we can see that our NASR pipeline is able to fix the majority of errors that are possible with this ontology. With a more sophisticated ontology, the improvement would be even greater.
>
> **Q(10): Additional baseline from [1] ROAD-R: The Autonomous Driving Dataset with Logical Requirements.**
>
> Thanks for the suggestion. We will cite this paper and have started to implement this baseline, but we are not able to complete it yet as there are lots of engineering implementation details to solve to adapt [1] to our problem setting. We are happy to update the result if we finish in time, or include it in the camera ready.
>
> Overall our understanding of [1] is that, given the output of a NN, it translates the problem of finding the minimal set of predictions to change to obtain a feasible solution to a weighted partial maximum satisfiability problem, and calls a PMaxSAT solver after inference. In this regard, it can indeed be applied to our experiments. However, we expect it will not perform any better than e.g, SatNET and NeurASP, etc, in that there is no module for “where to reason” (cf: our Mask-Predictor), and thus the symbolic part of the inference step is expected to be expensive.
>
> **C1: The expressivity of the constraints admitted by the framework. It seems to be it is propositional logic, as the pipeline requires to have one element in the prediction of the neuro-solver for each atom (this implies that even if you admit predicates, the space of your variables is still finite, thus the predicates are just syntactic sugar)**
>
> There is no limitation on the constraints admitted by the framework, as long as they can be dealt with by an available symbolic solver, which we treat as a subroutine. The choice of the symbolic solver should be done accordingly.
>
> Most importantly, the constraints need not necessarily even be logic constraints, they can also be of different nature such as physics rules.
>
> For example, suppose that the input of the system is a set of objects governed by Newtonian physics. We have a set of n objects and we want to predict what happens, after T seconds, if we apply a force of 5 Newton. Then the output of the neuro-solver consists of  n pairs of coordinates and velocity (one for each object) at time T. Here the mask-predictor identifies which predictions provide incorrect values for the objects’ coordinates and velocities. The symbolic solver can re-compute the values of coordinate and velocity for these objects only (which is computationally intensive) to obtain the correct ones. Here the constraints are differential equations from mechanics and the Symbolic-Solver is a differential equation solver. In this scenario the semantics of the Mask-Predictor is to verify that a set of coordinate-velocity pairs is the correct solution of a system of equations.
>
> The only limitation is that the number of outcome variables are finite. IE: The variables that define the “API” between the neuro-solver and the symbolic solver should be finite.  We do not impose any constraints on the form of the rules and/or additional facts (not in the output of the networks).
>
> Thus we do not restrict to (propositional) logic.
>
> Moreover, our pipeline can handle more than multi-label classification problems: for example, like in the physics scenario introduced above, the output of the neural network can be continuous values (i.e. the velocity/coordinates of the objects).

---

> ### Author Response · Authors · 2022-11-18
> **Rebuttal - part 3**
>
> **C2: The exact machine learning task on which the authors are trying to impose the constraints (e.g., supervised, semi-supervised etc.). It seems to be the fully-supervised setting. And in particular a case of a multilabel classification problem, where each output must be in [0,1] and there can be multiple labels associated with each prediction.**
>
> The experiments shown in the paper are in a fully supervised setting, so that could be the simplest way to describe our contribution. But we remark that this is not a strict requirement for the overall framework: (1) Each of the individual neural modules could be trained in a semi-supervised or unsupervised way. E.g., perception via unsupervised representation learning. (2) The whole pipeline can also be trained just using as reward the feedback from the symbolic-solver (e.g. if it finds a solution or not) without looking at the ground truth solution board. In this case it would be a true RL problem, as opposed to the current case which the reviewer correctly identifies as fully supervised, and where RL is merely used to get around the non-differentiability of the symbolic solver.
>
> **C3: Figure 1 and example 2 are misleading. The authors implement and test only for the sudoku task, and they do not go into the details on how the architecture would look like for the problem of object detection with constraints. Thus they should be removed.**
>
> The experiments shown in the paper are for the visual sudoku task, but the paper introduces a more general architecture that works for many different scenarios other than the visual sudoku one. The visual sudoku is just one specific show-case application for a more general idea.
>
> We added in the Appendix (see section A.8) some preliminary results on the SceneGraph domain (PredCl), as described in Example2 and Figure1. We considered a very simple ontology that can provide only a small improvement. However, we can see that the pipeline achieves good performance and is able to recover the majority of the error corrections possible. With a more sophisticated ontology, the improvement would be even greater.
>
> **C4: The paper states that as long as there are no false negatives the constraints are satisfied. This I think would apply only in the case of object detection (while in the case of the sudoku, everything should work even with false negatives. Is that correct?)**
>
> Let’s consider the label $1$ (=keep the prediction) as positive examples and $0$ (=remove the prediction) as negative examples (we adopted this convention in the paper to avoid confusion). In this case constraints might be able to be satisfied even with false negatives in the mask, assuming that the Hadamard product of the neuro-solver and the mask predictor output is satisfiable. However, it’s possible that if the neuro-solver outputs an inconsistent board, and it does not become satisfiable after the mask is applied, then the final prediction might not satisfy the constraints. Thus constraint satisfaction is only guaranteed in the case of no false positives by the MP.
>
> **C5: Figure 1 and 2 are misplaced. They should be moved earlier in the paper.**
>
> Thanks, we agree. Normally latex manages the placing. We’ll see if we can convince it to move them while meeting the space constraints.
>
> **C6: I personally would change the "Neuro-Solver" to simply neural-network. This would make the pipeline much clearer. (In the end these will literally be one or more neural networks).**
>
> We agree that all the modules (Neuro-Solver, SolverNN, Perception and Mask-Predictor), except the Symbolic-Solver, are all neural networks. However, they have completely different roles and learn different functions. For this reason they need to be distinguished with different names. Moreover, it is useful to be able to disentangle the impact of each one (e.g., in tab 3) without creating confusion between them.
>
> **C7: You introduce the "Perception" module only in the "Experimental Setup" section, which might be confusing for the reader. I would again just talk about "neural network" and then say that for the particular sudoku task you have one that takes care of the perception and one to fill in the sudoku.**
>
> We recap that the Neuro-Solver is the composition of the Perception and SolverNN. The Neuro-Solver has been introduced in Section2 when introducing the general pipeline outline. The Perception and the SolverNN are a specific implementation of the Neuro-Solver. However this is just one of the many possibilities: e.g. the Neuro-Solver can be just one big Transformer model that takes in input a Sudoku board image and outputs the solution directly.  Another example is the case of Predicate Classification task where the Neuro-Solver is a single architecture taking in input an image and the box and labels of the objects in it and outputting a set of relationship triples.
>
> **Q:  Code release?**
>
> Yes, we will be pleased to release the code upon acceptance.

---

> > ### Comment · Reviewer_ndKP · 2022-12-09
> > **Thank you for your replies**
> >
> > Sorry for the delayed reply and thanks for the detailed comments. All my concerned have been addressed.

---

### Official Review · Reviewer_8rGg · 2022-10-24

**Confidence:** 4
**Correctness:** 2
**Technical Novelty And Significance:** 3
**Empirical Novelty And Significance:** 3
**Recommendation:** 8

**Clarity, Quality, Novelty And Reproducibility:**

**Clarity**:  The paper is reasonably well written.  There are only a handful of minor issues with the text, including:
- p 4: "that do not violateS"
- p 5: "r indicates the reward" - not defined yet, please add a forward pointer to the equation in p. 7.
- p 6: "ADADELTA optimizer" -> the ADADELTA optimizer.
- p 6: "the number of output layer being 1-dimensional" - rephrase.
- p 7: "we improve the performance" -> improve on
- Equations: the $\odot$ symbol stands for Hadamard product, but it cannot
possibly work as intended if the deleted cells have value $0$, as shown in,
e.g., Figure 2.
These can all be easily fixed.

(1) One aspect that is not clear from the text is how (masked) neural predictions that are unsatisfiable are handled by the symbolic solver.  This information *might* be part of Table 3, which was however hard for me to parse.  One option would be to split the Table into several sub-tables and provide more hand-holding in its interpretation.

**Quality**: The proposed approach is sensible and appears to work as intended.  The idea of using attention to combine a neural and a symbolic solver is actually quite clever.  The empirical evaluation also looks solid, but incomplete, for two reasons.

(2)  NASR is only evaluated on visual sudoku.  This is an interesting application in that it is to complex for many exact NeSy approaches.  It is also true that the evaluation encompasses four different visual sudoku datasets.  However, given that NASR scales better than existing approaches, it would have been nice to apply it to more complex and realistic applications than sudoku alone.

(3) A more critical issue is that - as far as I could see - the training times are not reported.  (Did I miss anything?)  Training time is not a detail:  the *whole point* of NASR is to make inference more efficient at the expense of training time.  As long as the neural solver is task-specific (meaning that it has to be trained anew for each task being solved), training time cannot be trivially amortized across tasks, although it *can* be amortized across test points.  Now, training NASR involves combining supervised learning and an RL stage, which in turn requires to invoke the symbolic solver (correct?).  I expect this step to be quite time consuming, and actually more time consuming than training a neural solver alone (no RL).  The symbolic solver easily wins the competition, as it requires no training at all.  Of course I might be mistaken, and it could be that NASR is very efficient to train - but this should be clearly shown empirically.  A plot that compares training times of different approaches would be very useful - and I think quite important.  I expect the authors to address this point in their rebuttal.

I *will* increase my score once this issue is resolved.

(4) Sometimes NASR is just not worth the hassle.  Namely, if the neural reasoner performs poorly, NASR is essentially equivalent (in terms of accuracy) to but slightly slower (due to the less-than-useful neural step) than the symbolic reasoner.  See for instance Figure 7.  This is mentioned briefly in p. 2, but I think it deserves to be stated more clearly in the main text, at the bare minimum in the conclusion.  This is not a huge deal for me, and the issue of how to prevent this from happening could be addressed in future work.

(5) One final aspect is that it is not clear (to me, at least) how infeasible (masked) neural predictions are dealt with, as in this case the symbolic solver cannot provide any output.  How are these accounted for in the performance measures?

**Novelty**:  The proposed method is novel and clearly positioned against the state of the art.

A few methods (like ProbLog, Vampire, etc.) mentioned in the "Symbolic Solver" paragraph are used without references.  These should be added.

**Reproducibility**  The paper is not accompanied by source code and I could not find any mention of plans for future code releases.

As a side note, I am grateful that the authors didn't make use of the "thinking fast and slow" slogan.

**Strength And Weaknesses:**

PROS
- The method is significant in that it targets a known weakness of existing NeSy approaches.
- The proposed pipeline (and accompanying training procedure) is intuitively appealing.
- The empirical results on visual sudoku look promising.
- The text is generally well structured and easy to follow.
- The related work section is reasonable and gives an overall fair representation of existing approaches.

CONS
- The evaluation only considers visual sudoku.
- The results do not report the time required for training the various models.
- Improvement over the symbolic baseline can be small if the neural method makes many mistakes.
- Some references are missing.
- Some aspects of the pipeline are a bit unclear.

**Summary Of The Paper:**

The authors introduce a hybrid neuro-symbolic pipeline, denoted NASR, that combines a fast but inaccurate neural reasoner with an accurate but slow symbolic reasoner.  The two components are combined using a neural module trained to identify mistakes in the output of the neural reasoner, and using the symbolic reasoner to fill in the gaps.  The resulting pipeline is trained using a mixture of supervised learning and reinforcement learning.  Empirical results on four visual sudoku data sets with different characteristics are reported, showing improved accuracy over the neural reasoner at a much lower cost than a fully symbolic solution.

**Summary Of The Review:**

Intuitively appealing and promising contribution with flawed empirical evaluation.

---

> ### Author Response · Authors · 2022-11-18
> **Rebuttal - part 1**
>
> **Q1: Table 3 is hard to parse. One option would be to split the Table into several sub-tables and provide more hand-holding in its interpretation.**
>
> Sorry for the confusion. To clarify:
> The first 3 lines correspond to the performance of the supervised training of the single components in isolation.
> The middle block of 3 lines corresponds to the neural-solver part of the pipeline (perception+solverNN), and the two versions of the end-to-end pipeline (with or without RL). For Example, considering the dataset big_kaggle, the neuro-solver produces 47.03% of completely corrected boards. The remaining 52.97% can still be fixed by the MaskPredictor+SymbolicSolver part of the pipeline. This only happens when the mask predictor identifies all the wrong predictions. In the case of big_kaggle this happens for ~37% of boards.
> We tried to format the table differently in a way that is easier to parse. For each dataset we have two columns: the first one corresponds to the cell-wise metrics for the single neural components; the second column corresponds to the board-wise metric of “percentage” of correct boards.
>
> **Q2: NASR is only evaluated on visual sudoku (more complex and realistic applications [The evaluation only considers visual sudoku.]**
>
> We added in the Appendix (see section A.8) some preliminary results on the Scene Graph domain (PredCl), as described in Example 2 and Figure 1. Existing publicly available ontologies are very simple, so the maximum achievable improvement with, even by exhaustive reasoning, is quite small. However, we can see that our NASR pipeline is able to fix the majority of errors that are possible with this ontology. With a more sophisticated ontology, the improvement would be even greater.
>
> **Q3:   The training times are not reported.**
>
> We have now added a table in the Appendix (Section A.6, Table 4)  recording the training time of the different neural components as well as for the RL refinement.
>
> We would like to emphasize that we only claim to have an acceleration in testing  (inference) time, widely accepted as important in the ML community, not in the training time. As the reviewer mentioned, we expect the training time to be amortized across testing points, and training can be conducted offline in advance, so we are mainly  interested in the inference cost of the models in the online setting (Fig 3).
>
> Table 4 shows that training time of our model is indeed greater than the symbolic baseline (which requires training time of the perception sub-module alone), and that indeed RL takes up much of the time. But it is not dramatically greater than the symbolic baseline. The table also shows that other neuro-symbolic models such as SatNet also sometimes have noticeably higher training time than the symbolic baseline, and our framework is not an outlier with respect to these.
>
> **Q4: Sometimes NASR is just not worth the hassle. Improvement over the symbolic baseline can be small if the neural method makes many mistakes.**
>
> While there are some special cases (like minimal_17 dataset) in which we do not have a noticeable boost, for general boards our system provides an advantage over the symbolic baseline, both in performance and computational time. This makes NASR a good choice if the type of board is not known in advance. Moreover NASR is more robust to noise (see section A.2 in the appendix). For example, considering multiple_solution dataset, NASR without RL performs similarly to the symbolic baseline. However, considering the analysis in section A.2, NASR without RL is to be preferred to the symbolic baseline .
>
> **Q5: One aspect that is not clear from the text is how (masked) neural predictions that are unsatisfiable are handled by the symbolic solver.**
>
> When an unsatisfiable neural prediction is given in input to the symbolic solver the solver cannot provide an answer (it fails).  In this case no solution is produced and the whole board is counted as wrong in the evaluation. Thus, a board is counted as correct only if the neural component produces a satisfiable prediction, and all the resulting cells are correct. We do not reward partially correct (e.g., 90% cells correct) boards in the evaluation.
> We added a comment to disambiguate this in the paper.
>
> **Q6: A few methods (like ProbLog, Vampire, etc.) mentioned in the "Symbolic Solver" paragraph are used without references. These should be added.**
>
> Thanks. Done.

---

> > ### Comment · Reviewer_8rGg · 2022-11-19
> > **Reply**
> >
> > **Table 3 has been reworked.**  Thank you.
> >
> > **Scene Graph experiments.** I am quite impressed by how quickly this was set up.  The experiment is very reasonable and indicates that NASR *can* work in non-Sudoku domains.  The gains are not huge, but at least they are consistently there.
> >
> > **Training times.***  Thank you, Appendix A.6 answers satisfactorily my question and shows unequivocally that (for Sudoku) NASR is not dramatically slower than alternative approaches, which is what I was worried about.  I will increase my score accordingly.  (I am a bit surprised that perceptron takes so long to train, but this is a tiny detail.)
> >
> > Concerning the message: yes, I understand that the authors aim at improving inference time.  The reason I was worried is that if training time explodes, then NASR might not be as widely applicable.  This is something that readers *need* to know.  This turns out not to be the case, so this is all good.
> >
> > **Q4.*** Agreed.  This was not a major issue for me to begin with.  The answer to Q5 actually addresses this very point for me.
> >
> > **Handling of unsatisfiable partial predictions.** Thank you, the way performance is computed seems fair to me.  Since this had me unnecessarily worried, I will increase my score accordingly.

---

> ### Author Response · Authors · 2022-11-18
> **Rebuttal - part 2**
>
> **Clarity:  How does the Hadamard product work if deleted cells have value 0?**
>
> There is one Hadamard product (element wise product) between the output of the NS and the output of the MP. The resulting vector has a 0 corresponding to the variable that the symbolic solver needs to fix. In our experiments, the solver expects that a 0 value indicates an unknown variable to be filled.
> We made this more clear for the general case where the solver expects another symbol such as “?” to indicate an unknown by adding an adaptor function that translates the NN output into the solver input format around equation (1).
>
> **Clarity: Typos & editorial.**
>
> Thanks. We fixed them.

---

### Official Review · Reviewer_6T4W · 2022-10-24

**Confidence:** 4
**Correctness:** 3
**Technical Novelty And Significance:** 3
**Empirical Novelty And Significance:** 3
**Recommendation:** 6

**Clarity, Quality, Novelty And Reproducibility:**

Questions:
    (1) For the RL refinement step, both the NS and MP models' parameters are updated right?
    (2) For computing the accuracy, are you directly compare the answers and the labels?

Minor:
    (1) For fig 2, the two from the last digit of the ns solution, masked solution vector are not match.

**Details Of Ethics Concerns:**

N/A.

**Strength And Weaknesses:**

Strength:
    (1) Use reinforcement learning to combine the perception part and the reasoning part.
    (2) Predict the incorrect parts of the initial solution to reduce the search space for the logical solver.

Weaknesses:
    (1) It would be better to show how the framework on another domain, now there is only one single domain: Visual Sudoku. The predicate classification task is really a good one.
    (2) It would be better to give or show the inituion of why the framewoke works. From the experimental results, the framework works better on SATNET dataset (avg 36 hints) v.s. the 17hints dataset. That's very interesting since from the sudoku perspective, 17 hints is supposed to be much harder than 36 hints. But for here, this can be understand since 36 hints means more images ("noisy digits") that the model needs to recongnise. So here comes the question, is the framework learned which digits are classified incorrect or learned something else?
    (3) The input of the mask predictor is only the solution generated by the neural solver, following (2), then this mask predictor cannot consider the error of the perception results. So I am wondering if the mask predictor can also consider use the the original problem as part of the input?
    (4) Some literature are missing, e.g., [1][2][3]. Though you are focusing on different perspectives and the setting are not the same. But it would be better to talk about them in the paper.

[1] Brouard, Céline, Simon de Givry, and Thomas Schiex. "Pushing data into cp models using graphical model learning and solving." International Conference on Principles and Practice of Constraint Programming. Springer, Cham, 2020.
[2] Mulamba, Maxime, et al. "Hybrid classification and reasoning for image-based constraint solving." International Conference on Integration of Constraint Programming, Artificial Intelligence, and Operations Research. Springer, Cham, 2020.
[3] Bai, Yiwei, Di Chen, and Carla P. Gomes. "CLR-DRNets: Curriculum Learning with Restarts to Solve Visual Combinatorial Games." 27th International Conference on Principles and Practice of Constraint Programming (CP 2021). Schloss Dagstuhl-Leibniz-Zentrum für Informatik, 2021.

**Summary Of The Paper:**

The authors proposes a new framework that tries to combine neural perception and symbolic reasoning.  In general, this framework consists of three components: (1) Neural Solver (NS) (2) Mask Predictor (MP) and (3) Logical Solver (LS).  The NS computes an initial solution for the input tasks, then the MP predicts the incorrect parts of the initial solution and a masked solution (the incorrect parts are masked as 0) is generated. Fianlly, this masked solution is fed into the LS to get the final solution. The NS and MP models are trained with supervised learning firstly. Then the reinforcement learning (RL) is employed to fine-tune the NS and MP models leveraging the LS results. The authors validate their framework on Visual-Sudoku tasks and shows that in most cases, they can perform better than the state-of-the-art methods.

**Summary Of The Review:**

The paper proposes a novel neural-symbol framework. Its basic idea is to use neural network to compute an intial answer, and predict the incorrect parts of it then use a logical solver to get the final answer. However, the framework is only validated on a single domain: visual Sudoku.

---

> ### Author Response · Authors · 2022-11-18
> **Rebuttal**
>
> **Q1: It would be better to show how the framework on another domain**
>
> We added in the Appendix (see section A.8) some preliminary results on the Scene Graph domain (PredCl), as described in Example 2 and Figure 1. Existing publicly available ontologies are very simple, so the maximum achievable improvement with, even by exhaustive reasoning, is quite small. However, we can see that our NASR pipeline is able to fix the majority of errors that are possible with this ontology. With a more sophisticated ontology, the improvement would be even greater.
>
> **Q2: Intuition of why the framework works:  is the framework learned which digits are classified incorrect or learned something else?**
>
> Overall the framework (SolverNN+Mask-Predictor+Reasoning) learns both to correct incorrectly classified digits, and also to infer missing digits. In summary, SolverNN guesses a complete solution to the board given the hints, mask-predictor performs consistency checking to identify which cells of this solution are likely to be incorrect, and reasoning fills in the cells identified by mask-predictor as potential errors.
> We added a section (Section A.7 in the appendix) with some experiments to disentangle the roles of the Perception, the SolverNN and the Mask-Prediction in the pipeline.
> The analysis shows that perception errors are almost all already corrected by SolverNN, and that errors remaining after SolverNN are further identified by Mask-Predictor (and thus made available for correction by Reasoning).
> The analysis also confirms that Mask-Predictor can correct errors both in the hints (perception errors) and in the solutions (SolverNN inference errors). It is not specific to one or the other.
>
> **Q3: The mask predictor can also consider use the the original problem as part of the input?**
>
> Thanks for the suggestion. Currently the Mask-Predictor considers the original problem indirectly (through the output of the Neuro-Solver) when refined with RL.
> It could be possible to provide the raw input image as part of the input of the Mask-Predictor. However, this would make the task of Mask-Predictor much more difficult, requiring it to also learn perception and board completion. We conjecture that this would probably hinder rather than help learning,  since it would make the task to learn by the Mask-Predictor more difficult (see also answer to Reviewer1-PAJH).
> Another possibility would be to provide both the Perception and the SolverNN output to the Mask-Predictor (rather than SolverNN output alone). We conjecture that this could facilitate the Mask-Predictor’s task by providing evidence on which cells are likely perceived and which are deduced. However, unfortunately we did not have time to complete this experiment in the rebuttal window.
>
> **Q4: Some literature are missing, e.g., [1][2][3].**
>
> Thanks. We added the missing references to the paper.
>
> **Q5: For the RL refinement step, both the NS and MP models' parameters are updated right?**
>
> Yes. However it is possible to use RL to refine the networks (NS and MP) separately if desired.
>
> **Q6: For computing the accuracy, are you directly compare the answers and the labels?**
>
> Yes, we compare with the ground truth labels. A board is counted as correct only if all the cells are correct. For NASR pipeline evaluation we do not reward partially correct boards (e.g. 90% of the cells are correct).
> In the case of multiple_solution dataset we only consider one solution per input board (the smallest in the lexicographical order). For testing and evaluation we consider all the solutions (ie: any valid solution is considered correct - so as not to disadvantage the performance evaluations). However, considering only the smallest solution (in the lexicographical order) does not change the performance significantly.
>
> **Q7:  For fig 2, the two from the last digit of the ns solution, masked solution vector are not match.**
>
> Thanks for noticing, we fixed it.

---

### Official Review · Reviewer_PAJH · 2022-10-25

**Confidence:** 4
**Correctness:** 4
**Technical Novelty And Significance:** 4
**Empirical Novelty And Significance:** 4
**Recommendation:** 8

**Clarity, Quality, Novelty And Reproducibility:**

The paper is clearly written, and the proposal is novel. The authors do not provide code for the reported simulations.

**Strength And Weaknesses:**

## Strengths
- The proposed method presents an interesting and elegant way of combining the relative strengths of neural and symbolic reasoning approaches. The use of a learned module to decide when to invoke the symbolic reasoning module is a nice solution that avoids the circular challenge of using a symbolic module to check whether the neural solver is correct, and the approach works surprisingly well (one might have expected that the learned mask-predictor would suffer from the same limitations that the neural solver does in the first place).
- The method performs well against other neurosymbolic approaches (and against purely neural or purely symbolic approaches) on a challenging visual sudoko task.
- The model is robust to perceptual noise, and can solve problems in an efficient manner.

## Weaknesses
- The major weakness is that the approach still requires the use of handcoded, task-specific knowledge, i.e. the rules of sudoku are programmed into the symbolic module, rather than being able to learn the rules or receive the rules in natural language as human reasoners do. However, despite this limitation (which is broadly shared by current neurosymbolic methods, and a major open challenge), the proposed method constitutes an elegant solution to a specific problem in integrating neural and symbolic approaches, and should be a useful contribution to the literature.

### Other notes
- There is an interesting connection between the proposed approach and the 'two systems' perspective in psychology (e.g. [1] and [2]).

[1] Kahneman, D. (2011). Thinking, fast and slow.

[2] Sloman, S. A. (1996). The empirical case for two systems of reasoning. Psychological bulletin, 119(1), 3.


**Summary Of The Paper:**

This paper proposes a novel neurosymbolic method that allows an efficient (but potentially error-prone) neural solver to be corrected by a precise (but inefficient) symbolic module. The intervention from the symbolic module is mediated by a learned neural module, allowing the overall system to exploit both the efficiency of the neural solver, and the precision of the symbolic solver only when needed.

**Summary Of The Review:**

The paper proposes an interesting neurosymbolic method that can exploit both the efficiency of a neural reasoning approach together with the precision of a symbolic reasoning approach, mediated by a neural module that learns when to invoke the symbolic module. The method elegantly optimizes the tradeoff between efficiency and precision, and outperforms competing neurosymbolic methods.

---

> ### Author Response · Authors · 2022-11-18
> **Rebuttal**
>
> **Q: The approach works surprisingly well (one might have expected that the learned mask-predictor would suffer from the same limitations that the neural solver does in the first place)**
>
> Thanks for acknowledging how well our approach works. The explanation is that the mask-predictor needs to learn an easier task compared to the neuro-solver: The neuro-solver needs to be able to process raw input as well as to solve the task, while the mask-predictor just needs to identify the errors. The task of the mask-predictor is then a sort of “consistency” checking over the solution, which is much easier compared to the full solution generation.
>
> **Q: The major weakness is that the approach still requires the use of handcoded, task-specific knowledge**
>
> We agree that is a valid point and that an ideal tool would also be able to learn the rules and exploit them at the same time. However, rule induction comes with its own set of different limitations. Thus both rule induction and rule exploitation approaches are distinct lines of research that both have value to the community.
>
> **Q: There is an interesting connection between the proposed approach and the 'two systems' perspective in psychology (e.g. [1] and [2]).**
>
> Thanks. We agree. We added references in the paper to both of these works, as well as to LeCun’s paper (A Path Towards Autonomous Machine Intelligence) and Booch’s paper (Thinking Fast and Slow in AI).

---

> > ### Comment · Reviewer_PAJH · 2022-11-19
> > **Reply**
> >
> > Thanks very much to the authors for this reply, I appreciate the additional discussion and comments. The explanation re: why the mask predictor works is informative, and these additional references in the intro make for some nice connections to the literature.

---

### Author Response · Authors · 2022-11-18
**Summary of revisions**

Dear AC and reviewers,

We sincerely thank AC and all reviewers’ time and efforts in reviewing our paper. The constructive suggestions have helped us to improve our paper further. We appreciate that reviewers find that our idea is “novel and clearly positioned”,  “interesting and elegant”, “intuitively appealing” and with “strong empirical results”.

The major queries raised by the reviewers, and their resolution, are summarized as follows:
* **Reporting training time (8rGg):** Our main focus is on achieving fast inference time. However, for completeness, we now report the training time in Appendix A6, which shows that our NASR is comparable to the competitors.
* **Other tasks (All reviewers):** In addition to our results on visual Sudoku, we now present results on predicate classification in the visual  scene graph domain. The results in Appendix A8 show that NASR achieves most of the maximum achievable benefit of symbolic reasoning given a simple available domain-range ontology.
* **More intuition (6T4W):** To give additional intuition into the role of each component, we now also report a breakdown of statistics on which types of errors were corrected by which component of the system in Appendix A7.

We believe  that our revisions and clarifications address all the reviewers’ queries. We are happy to answer any further questions that may arise.

---

### Decision · Program_Chairs · 2023-01-20

**Decision:**

Accept: notable-top-5%

**Justification For Why Not Higher Score:**

N/A

**Justification For Why Not Lower Score:**

The paper gets high scores from 3 reviewers who found the approach interesting. The technique is intuitive and could be easily explained in an oral presentation. The subject mater is timely and of interest to the community.

**Metareview: Summary, Strengths And Weaknesses:**

The paper describes a novel pipeline for combining neural and symbolic modules. A mask-predictor is used to identify potential mistakes in the predictions of the neural module, and the symbolic module is invoked to fix these mistakes.

All reviewers recommend accepting the paper. They appreciate the technique, deeming it novel, elegant and intuitive. The empirical evaluation shows that the proposed method performs well. The major weakness that the reviewers identified is that the technique was evaluated on a single domain. During the discussion period the authors took steps to alleviate this weakness by providing experiments on a second domain and showing consistent, albeit small improvements.

**Note From Pc:**

if the above contains the word "oral" or "spotlight" please see: "oral" presentation means -> notable-top-5% and "spotlight" means -> notable-top-25%. As stated in our emails, we are disassociating presentation type from AC recommendations